# Doubly Robust Off-Policy Value and Gradient Estimation for Deterministic Policies

**Nathan Kallus, Masatoshi Uehara** *
Cornell University and Cornell Tech
New York, NY
kallus@cornell.edu, mu223@cornell.edu

## Abstract

Offline reinforcement learning, wherein one uses off-policy data logged by a fixed behavior policy to evaluate and learn new policies, is crucial in applications where experimentation is limited such as medicine. We study the estimation of policy value and gradient of a deterministic policy from off-policy data when actions are continuous. Targeting deterministic policies, for which action is a deterministic function of state, is crucial since optimal policies are always deterministic (up to ties). In this setting, standard importance sampling and doubly robust estimators for policy value and gradient fail because the density ratio does not exist. To circumvent this issue, we propose several new doubly robust estimators based on different kernelization approaches. We analyze the asymptotic mean-squared error of each of these under mild rate conditions for nuisance estimators. Specifically, we demonstrate how to obtain a rate that is independent of the horizon length.

## 1   Introduction

Offline reinforcement learning (RL), wherein one uses off-policy data logged by a fixed behavior policy to evaluate and learn new policies, is crucial in applications where experimentation is limited (Farajtabar et al., 2018; Bibaut et al., 2019; Liu et al., 2018; Kallus and Uehara, 2019b). A key application is RL for healthcare (Gottesman et al., 2019; Murphy, 2003). Since it is not possible to collect new data, it is crucial to efficiently use the available data. Recent work on off-policy evaluation (OPE; Kallus and Uehara, 2020a, 2019a) have shown how efficiently taking advantage of problem structure, such as Markovianness and ergodicity, can improve OPE and tackle the well-known issue of OPE known as the curse of horizon (Liu et al., 2018). Kallus and Uehara (2020b) applied these advances to off-policy *learning* using a policy gradient approach, i.e., proposing efficient off-policy estimators for the policy gradient and incorporating them into gradient ascent methods.

All the aforementioned methods, however, cannot be directly applied to the evaluation and learning of deterministic policies when actions are continuous since the density ratio (Radon-Nikodym derivative) does not exist: the behavior policy usually has zero mass on single actions (more generally, it can have at most countably-many atoms while the evaluation policy may take any of a continuum of actions). This question is important since the maximum-value policy is generally deterministic (up to ties between actions) so if one seeks optimal policies one should focus on deterministic ones. In the bandit setting (horizon of one action), several recent works tackle this problem (Bibaut and J. van der Laan, 2017; Kallus and Zhou, 2018; Colangelo and Lee, 2019), but applying these methods in a straightforward manner to RL may lead to a bad convergence rate that deteriorates with horizon.

In this paper, we propose several doubly robust off-policy value and gradient estimators for deterministic policies in an RL setting. We analyze the asymptotic mean-squared error (MSE) of each

---

Table 1: Comparison of off-policy value and gradient estimators for deterministic policies. Proposed estimators are typeset in bold. "MSE" is the *convergence rate* when nuisances are estimated at rate under "Rate," *irrespective* of choice of nuisance estimators. "–" means the convergence rate depends on the choice of estimators. "Nuis" are nuisances for OPE. "Nuis+" are *additional* nuisances for policy gradient. "D/I" is whether differentiation (D) or integration (I) of $\hat{q}_t(s_t, a_t)$ wrt $a_t$ is required.

| | (a) OPE | | | | (b) Off-policy gradient | | | |
|---|---|---|---|---|---|---|---|---|
| | MSE | Nuis | Rate | | MSE | Nuis+ | Rate | D/I |
| DM | – | $q^{\pi^e}$ | – | DPG | – | $d^{q^{\pi^e}}$ | – | D |
| **CDRD** | $n^{-\frac{4}{H+4}}$ | $\lambda^{\mathcal{K}}, q^{\mathcal{K}}$ | $n^{-\frac{1}{H+4}}$ | **CPGD** | $n^{-\frac{4}{H+6}}$ | $d^{\lambda^{\mathcal{K}}}, d^{q^{\mathcal{K}}}$ | $n^{-\frac{1}{H+6}}$ | D |
| **CDRK** | $n^{-\frac{4}{H+4}}$ | $\lambda^{\pi^e}, q^{\pi^e}$ | $n^{-\frac{1}{H+4}}$ | **CPGK** | $n^{-\frac{4}{H+6}}$ | $d^{\lambda^{\pi^e}}, d^{q^{\pi^e}}$ | $n^{-\frac{1}{H+6}}$ | I |
| **MDRD** | $n^{-\frac{4}{5}}$ | $w^{\mathcal{K}}, q^{\mathcal{K}}$ | $n^{-\frac{1}{5}}$ | **MPGD** | $n^{-\frac{4}{7}}$ | $d^{w^{\mathcal{K}}}, d^{q^{\mathcal{K}}}$ | $n^{-\frac{1}{7}}$ | D |
| **MDRK** | $n^{-\frac{4}{5}}$ | $w^{\pi^e}, q^{\pi^e}$ | $n^{-\frac{1}{5}}$ | **MPGK** | $n^{-\frac{4}{7}}$ | $d^{w^{\pi^e}}, d^{q^{\pi^e}}$ | $n^{-\frac{1}{7}}$ | I |

estimator under extremely lax conditions that accommodate flexible learning of the nuisances that appear in the estimator (such as $q$-functions). Specifically, we propose estimators of policy value and gradient with MSE convergence rate that does not deteriorate with horizon and the leading term's coefficient has only a polynomial dependence on horizon. These results are summarized in Table 1.

## 2  Preliminaries

**Problem set up**   Consider an $H$-long time-varying Markov decision process (MDP), with states $s_t \in \mathcal{S}_t$, actions $a_t \in \mathcal{A}_t$, rewards $r_t \in [0, R_{\max}]$, initial state distribution $p_1(s_1)$, transition distributions $p_{t+1}(s_{t+1} \mid s_t, a_t)$, and reward distribution $p_t(r_t \mid s_t, a_t)$, for $t = 1, \ldots, H$. A policy $\pi = (\pi_t(a_t \mid s_t))_{t \leq H}$ induces a distribution over trajectories $\mathcal{T} = (s_1, a_1, r_1, \ldots, s_T, a_H, r_H)$:

$$p_\pi(\mathcal{T}) = p_1(s_1)\pi_1(a_1 \mid s_1)p_1(r_1 \mid s_1, a_1) \prod_{t=2}^{H} p_t(s_t \mid s_{t-1}, a_{t-1})\pi_t(a_t \mid s_t)p_t(r_t \mid s_t, a_t). \quad (1)$$

In this paper, we focus on *continuous* actions, $\mathcal{A}_t \subseteq \mathbb{R}$. For brevity we focus on the univariate case; the extension to multivariate actions is straightforward. We are interested in the value, $J = \mathbb{E}_{p_{\pi^e}}[\sum_{t=1}^{H} r_t]$, of a given policy, $\pi^e$, called the evaluation policy. In particular, we consider the case where $\pi^e$ is *deterministic* in that it is given by maps $\tau = (\tau_t)_{t \leq H}$, $\tau_t : \mathcal{S}_t \to \mathbb{R}$ such that $\pi_t^e(a_t \mid s_t) = \delta(a_t - \tau_t(s_t))$ is the Dirac measure at $\tau_t(s_t)$, meaning when we follow $\pi^e$, $a_t$ is a function $\tau_t$ of $s_t$. When $\tau$ is parametrized as $\tau = \tau_\theta$ by some parameter $\theta \in \Theta$, we are also often interested in the policy gradient, $Z = \nabla_\theta J$, as it can be used for policy *learning* via gradient ascent. We often drop the subscript and understand $\nabla$ to be with respect to (wrt) $\theta$. When studying policy gradient estimation, we will assume throughout that $\tau_{\theta,t}(s_t)$ is almost surely differentiable in $\theta$ and that $\|\nabla\tau_t(s_t)\|_{\mathrm{op}} \leq \Upsilon$ for some $\Upsilon < \infty$, where $\|\cdot\|_{\mathrm{op}}$ is the matrix operator norm. Additionally, in theoretical results, we will assume actions are bounded: $\mathrm{support}(\pi_t^b(\cdot \mid s_t)) = [0, 1]$, $\tau_t(s_t) \in (0, 1)$.

In the *offline* setting, the data available to us for estimating $J$ and $Z$ consists only of trajectory observations from some *different* fixed policy, $\pi^b$, called the *behavior policy*:

$$\mathcal{T}^{\langle 1 \rangle}, \ldots, \mathcal{T}^{\langle n \rangle} \sim p_{\pi^b}, \ \mathcal{T}^{\langle i \rangle} = (S_1^{\langle i \rangle}, A_1^{\langle i \rangle}, R_1^{\langle i \rangle}, \cdots, S_H^{\langle i \rangle}, A_H^{\langle i \rangle}, R_H^{\langle i \rangle}). \qquad \text{(off-policy data)}$$

Let $w_t^{\pi^e}(s_t) = p_{\pi^e}(s_t)/p_{\pi^b}(s_t)$ denote the marginal density ratio, where $p_{\pi^e}(s_t)$, $p_{\pi^b}(s_t)$ are the marginal densities of $s_t$ under $p_{\pi^e}(\mathcal{T})$, $p_{\pi^b}(\mathcal{T})$, respectively. Let $q_t^{\pi^e}(s_t, a_t) = \mathbb{E}_{p_{\pi^e}}[\sum_{k=t}^{H} r_t \mid s_t, a_t]$, $v_t^{\pi^e}(s_t) = \mathbb{E}_{p_{\pi^e}}[\sum_{k=t}^{H} r_t \mid s_t]$ be $\pi^e$'s $q$- and $v$-functions. We assume throughout that $1/\pi_t^b(a_t \mid s_t) \leq C_1$, $w_t^{\pi^e}(s_t) \leq C_2$. For clarity, we reserve capital letters for observed data, dropping superscripts $^{\langle i \rangle}$ for a generic data point, and use lower case for generic MDP random variables. All expectations without subscripts are taken wrt $p_\pi^b$. The *empirical expectation* is $\mathbb{P}_n f = \frac{1}{n}\sum_{i=1}^{n} f(\mathcal{T}^{\langle i \rangle})$. Define the $L_2$ norm as $\|f(\mathcal{T})\|_2 = \mathbb{E}[f(\mathcal{T}^2)]^{1/2}$. We let $\cdot^{(i)}$ denote the $i$-th order derivative wrt action alone and define the max Sobolev norm as $\|f\|_{j,\infty} = \max_{i=0,\cdots,j} \|f^{(j)}\|_\infty$. We emphasize that, e.g., $q^{(i)}(s_t, a_t)$ refers to differentiating only wrt $a_t$ alone.

**Integral kernels** In developing estimators we will use a second-order kernel $k : \mathbb{R} \to \mathbb{R}$, i.e., $\int k(u)\mathrm{d}u = 1$, $\int uk(u)\mathrm{d}u = 0$, $M_2(k) = \int u^2 k(u)\mathrm{d}u < \infty$. We define $\Omega_2^{(i)}(k) = \int (k^{(i)}(u))^2 \mathrm{d}u$. Given a bandwidth $h$, let $K_h(u) = h^{-1} k(u/h)$. Examples of differentiable kernels include Gaussian $k(u) \propto \exp(-u^2)$, biweight $k(u) \propto \max(0, 1-u^2)^2$, and triweight $k(u) \propto \max(0, 1-u^2)^3$.

**Background on Offline Evaluation and Policy Gradient** Direct estimation of $q$-functions (direct method, DM; Munos and Szepesvári, 2008) and step-wise importance sampling (IS; Precup et al., 2000) are two common approaches for OPE. However, the former is known to suffer from the high variance and the latter from model misspecification. The doubly robust (DR) estimate combines the two, but its asymptotic MSE can still grow exponentially in horizon (Jiang and Li, 2016; Thomas and Brunskill, 2016). Kallus and Uehara (2020a) show that the efficient MSE in the MDP case is polynomial in $\mathcal{O}(H^2/n)$ and give an estimator achieving it by combining marginalized IS (Xie et al., 2019) and $q$-modeling using cross-fold estimation (Chernozhukov et al., 2018). OPE and off-policy policy gradient estimation are closely related (Huang and Jiang, 2019). Efficiency analysis and efficient estimators for off-policy policy gradients was recently given in Kallus and Uehara (2020b).

All of the aforementioned methods assume that the density ratio $\pi_t^{\mathrm{e}}(a_t \mid s_t)/\pi_t^{\mathrm{b}}(a_t \mid s_t)$ exists and is bounded. However, when $\pi^{\mathrm{e}}$ is deterministic this is generally violated as it requires that $\pi_t^{\mathrm{b}}(a_t \mid s_t)$ has an atom at $\tau_t(s_t)$ but $\pi^{\mathrm{b}}$ usually has no atoms and at most can only have countably-many, while $\tau_t(s_t)$ can vary continuously, especially for policy learning (e.g., via gradient ascent). In the bandit setting ($H = 1$), Kallus and Zhou (2018) recently showed that OPE is feasible under additional smoothness assumptions on induced action densities. The core idea is to approximate the deterministic policy by stochastic policy based on kernels. Namely, under appropriate smoothness,

$$\lim_{h \to 0} \mathbb{E}\left[\frac{K_h(A_1 - \tau_1(S_1))R_1}{\pi^{\mathrm{b}}(A_1 | S_1)}\right] = \mathbb{E}_{\pi^{\mathrm{e}}}[r_1]. \tag{2}$$

Kallus and Zhou (2018), assuming known behavior policy, therefore propose an IS-type estimator $\mathbb{P}_n[\frac{K_h(A_1 - \tau_1(S_1))R_1}{\pi^{\mathrm{b}}(A_1 | S_1)}]$ for $h$ appropriately shrinking in $n$ and analyze its bias and variance. This essentially amounts to approximating the deterministic policy with a stochastic one concentrated near, but not fully at, $\tau_1(S_1)$. Several works similarly deal with causal inference with continuous treatments (Imai and van Dyk, 2004; Hirano and Imbens, 2005; Galvao and Wang, 2015; Fong et al., 2018; Wu et al., 2018; Kennedy et al., 2017; Su et al., 2019) focusing on a single ($H = 1$) constant ($\tau_1(s_1) = a_1^*$) action. Although we do not explicitly use counterfactual notation, our estimand is equivalent to a counterfactual one under sequential ignorability (Hernan and Robins, 2019).

## 3 Bandit Setting, $H = 1$

For lucid presentation, we first develop off-policy value and gradient estimators in the bandit setting, i.e., $H = 1$. In this section, since $H = 1$, we omit the time index, e.g., letting $S = S_1$, $q = q_1$, etc.

### 3.1 Off-Policy Deterministic Policy Evaluation

There are actually several different ways to kernelize the deterministic policy. To motivate our estimators, note that, under appropriate smoothness (see Theorem 1 below) we have that

$$\lim_{h \to 0} \mathbb{E}\left[\frac{K_h(A - \tau(S))\{R - f_1\}}{f_2} + f_3\right] = \mathbb{E}_{\pi^{\mathrm{e}}}[r_1], \tag{3}$$

when each of $f_1, f_2, f_3$ takes any of the following two values, giving $2^3$ possible combinations:

$$f_1 = \begin{cases} q(S, A) \\ q(S, \tau(S)) \end{cases}, \quad f_2 = \begin{cases} \pi^{\mathrm{b}}(A \mid S) \\ \pi^{\mathrm{b}}(\tau(S) \mid S) \end{cases}, \quad f_3 = \begin{cases} \int q(S, a) K_h(a - \tau(S))\mathrm{d}a \\ q(S, \tau(S)) \end{cases}.$$

This general form includes the form of some previously proposed estimators, including the IS-type estimator in Kallus and Zhou (2018) ($f_1 = 0$, $f_2 = \pi^{\mathrm{b}}(A \mid S)$, $f_3 = 0$), the DR-type estimator in Kallus and Zhou (2018) ($f_1 = q(S, A)$, $f_2 = \pi^{\mathrm{b}}(A \mid S)$, $f_3 = q(S, \tau(S))$), and the DR-type estimator in Colangelo and Lee (2019) ($f_1 = q(S, \tau(S))$, $f_2 = \pi^{\mathrm{b}}(\tau(S) \mid S)$, $f_3 = q(S, \tau(S))$).

Based on this, we propose a general DR-type estimator with nuisance functions $f_1, f_2, f_3$: first, we split the data randomly into two halves $\mathcal{U}_1$ and $\mathcal{U}_2$; then, we define the estimator as

$$\frac{1}{2}\mathbb{P}_{\mathcal{U}_1}\left[\frac{K_h(A - \tau(S))\{R - \hat{f}_1^{[1]}\}}{\hat{f}_2^{[1]}} + \hat{f}_3^{[1]}\right] + \frac{1}{2}\mathbb{P}_{\mathcal{U}_2}\left[\frac{K_h(A - \tau(S))\{R - \hat{f}_1^{[2]}\}}{\hat{f}_2^{[2]}} + \hat{f}_3^{[2]}\right], \tag{4}$$

where $\hat{f}_j^{[k]}$ is an estimate of $f_j$ based on estimating $\pi^b, q$ by $\hat{\pi}^{b,[k]}, \hat{q}^{[k]}$ using only the data in $\mathcal{U}_{3-k}$. This technique is called a cross-fitting, which is used to avoid metric entropy conditions on nuisance estimators (Chernozhukov et al., 2018). All of our nuisances can be estimated by standard nonparametric density or regression estimators (Hansen, 2009). To simplify notation, we often write this and similar estimators as $\hat{J} = \mathbb{P}_n \left[ K_h(A - \tau(S))\{R - \hat{f}_1\}/\hat{f}_2 + \hat{f}_3 \right]$, where implicitly $\hat{f}_j$ are fit using cross-fitting on the half of the data that excludes the data point on which it is evaluated.

We next analyze two primary cases: $f_1 = q(S, A)$, $f_2 = \pi^b(A \mid S)$, $f_3 = \int q(S, a)K_h(a - \tau(S))\mathrm{d}a$, which refer to as $\hat{J}^{\mathcal{K}}$ (for "kernel"), and $f_1 = q(S, \tau(S))$, $f_2 = \pi^b(\tau(S) \mid S)$, $f_3 = q(S, \tau(S))$, which refer to as $\hat{J}^{\mathcal{D}}$ (for "deterministic" as we plug in the deterministic policy into the nuisances).

**Theorem 1.** *Suppose for $i = 1, 2$, $\mathbb{E}[\|\hat{\pi}^{b,[i]} - \pi^b\|_{1,\infty}] = o(1)$, $\mathbb{E}[\|\hat{q}^{[i]} - q\|_{1,\infty}] = o(1)$, $\mathbb{E}[\|\hat{\pi}^{b,[i]} - \pi^b\|_\infty \|\hat{q}^{[i]} - q\|_\infty] = o((nh)^{-1/2})$, $nh^5 = \mathcal{O}(1)$, $nh \to \infty$, that $\pi^b(a \mid s)$, $q(s, a)$ are twice continuously differentiable wrt $a$ for almost all $s$, and that $\hat{\pi}^{b,[i]}, \hat{q}^{[i]}$ are uniformly bounded by a constant. Then, the bias and variance of $\hat{J}^{\mathcal{D}}$ are $\mathbb{E}[\hat{J}^{\mathcal{D}}] - J = 0.5M_2(k)h^2 B + o((nh)^{-1/2})$, $\mathrm{var}[\hat{J}^{\mathcal{D}}] = \frac{\Omega_2(k)}{nh}(V + o(1))$, where*

$$B = \mathbb{E}[q^{(2)}(S, \tau(S)) + 2q^{(1)}(S, \tau(S))\pi^{b(1)}(\tau(S)|S)/\pi^b(\tau(S)|S)], \quad V = \mathbb{E}\left[\frac{\mathrm{var}[R|S,\tau(S)]}{\pi^b(\tau(S)|S)}\right].$$

*If additionally $\hat{\pi}^{b,[i]}(a \mid s)$, $\hat{q}^{[i]}(s, a)$ are twice continuously differentiable wrt $a$, then the same holds for $\hat{J}^{\mathcal{K}}$ with $B = \mathbb{E}[q^{(2)}(S, \tau(S))]$ and the same $V$ as the above. In both cases, setting $h = \Theta(n^{-1/5})$ yields the minimal MSE of order $\mathcal{O}(n^{-4/5})$.*

Here, we assume that nuisance estimation errors converge in expectation (i.e., in $L_1$). If we instead assume the weaker convergence in probability, we can obtain the same guarantee on the bias and variance, conditioned on an event that occurs with high probability. Refer to Appendix E.

**Remark 1.** Three things should be noted. First, the best MSE rate achievable in the result is $\mathcal{O}(n^{-4/5})$, which is slower than the usual rate $\mathcal{O}(n^{-1})$ in OPE of stochastic policies under positivity. This slow rate is expected because our estimand in the deterministic case is not regular (Kennedy et al., 2017). Since the minimax rate for density estimation in a Sobolev class of smoothness parameter 2 is $\mathcal{O}(n^{-4/5})$ (Korostelev, 2011), we expect that this rate is minimax for $J$ among problems satisfying the conditions of Theorem 1. Establishing this formally is future work. Second, for $\hat{J}^{\mathcal{D}}$, the only condition on nuisance estimators is a sub-parametric rate, which can, e.g., be satisfied when each nuisance converges at the rate $o(n^{-1/5})$ for $h = \Theta(n^{-1/5})$. This condition appears weaker than the $o(n^{-1/4})$ nuisance rate required in usual OPE (Kallus and Uehara, 2020a; Chernozhukov et al., 2018); however, the required norm itself is stronger since $\|\cdot\|_2 \le \|\cdot\|_\infty$. In contrast, when just $q$ is estimated at rate $o(n^{-1/5})$, one cannot guarantee a similar $\mathcal{O}(n^{-4/5})$ MSE rate for DM without additional assumptions; a simple triangle inequality yields only an $o(n^{-2/5})$ MSE rate. The same is true for IS when we estimate $\pi^b$ only at $o(n^{-1/5})$ rate.[2] Third, both the variances and rates for $\hat{J}^{\mathcal{D}}$ and $\hat{J}^{\mathcal{K}}$ are the same, while the bias constant is slightly different. For brevity, in the following, we focus on deriving theoretical properties for $\hat{J}^{\mathcal{K}}$ where similar results are easily obtainable for $\hat{J}^{\mathcal{D}}$.

**Optimality of $\hat{J}^{\mathcal{K}}$ in terms of leading constant**   Kallus and Zhou (2018) computed the asymptotic bias and variance for the IS estimator ($f_1 = 0$, $f_2 = \pi^b(A \mid S)$, $f_3 = 0$) with known behavior policy (hence no nuisances). The estimator $\hat{J}^{\mathcal{K}}$ is obtained by adding the control variate $f_1 = f_3 = q(S, A)$ and while the two estimators have the *same* asymptotic bias, the leading variance term is *smaller*, having $\mathbb{E}\left[\frac{\mathrm{var}[R|S,\tau(S)]}{\pi^b(\tau(S)|S)}\right]$ instead of the larger $\mathbb{E}\left[\frac{\mathbb{E}[R^2|S,\tau(S)]}{\pi^b(\tau(S)|S)}\right]$ in Kallus and Zhou (2018). Thus, the advantage of our DR-type estimators over the IS estimator is not only ensuring $\mathcal{O}(n^{-4/5})$ convergence with an *unknown* behavior policy but also in providing an improvement in the variance leading term. In fact, we can prove $\hat{J}^{\mathcal{K}}$ is optimal among a class of estimators.

**Corollary 1.** *Assume $\pi^b(a|s), f(s, a), q(s, a)$ are $C^2$-functions wrt $a$. Then $\mathbb{P}_n[\frac{K_h(A - \tau(S))}{\pi^b(A|S)}\{R - f(S, A)\} + f(S, A)]$ has a bias independent of $f$ and variance minimized by letting $f = q$.*

This would correspond to an efficiency result in semiparametric theory, but that cannot applied here since our estimand is not regular (Kennedy et al., 2017). It is also difficult to compare $\hat{J}^{\mathcal{K}}$ and $\hat{J}^{\mathcal{D}}$ since the bias terms are different. We leave further investigation of optimality to future work.

**Remark 2** (Relation with previous literature)**.** Bibaut and J. van der Laan (2017) proposed a general approach for the estimation of non-regular estimands using smoothing. See the estimator in Example 3; however, they assumed a behavior policy is known. Foster and Syrgkanis (2019, Section 8) also touched on the idea of case $\mathcal{K}$. However, they did not analyze a mathematical detail.

### 3.2 Off-Policy Deterministic Policy Gradient Estimation

For a deterministic policy class $\{\tau_\theta(s) : \theta \in \Theta\}$, consider estimating $Z = \nabla J$ at a given $\theta$. Usually policy gradients involve the policy score, $\nabla \log(\pi_\theta^e(a \mid s))$ (Peters and Schaal, 2006; Kallus and Uehara, 2020b). However, for deterministic policies, these policy scores do not exist. However, assuming that $q$ is differentiable in $a$ immediately yields $Z = \mathbb{E}[q^{(1)}(S, \tau_\theta(S))\nabla\tau_\theta(S)]$, suggesting this may still be possible under appropriate smoothness.

**Deterministic Policy Gradient (DPG) and IS Policy Gradient (ISPG)** By taking a derivative of the IS and DM estimators wrt $\theta$, we obtain corresponding policy gradient estimators:

$$\hat{Z}^{\text{IS}} = \mathbb{P}_n[K_h^{(1)}(A - \tau(S))R/\hat{\pi}^b(A \mid S)], \quad \hat{Z}^{\text{DPG}} = \mathbb{P}_n[\hat{q}^{(1)}(S, \tau_\theta(S))\nabla_\theta\tau_\theta(S)], \quad (5)$$

where $K_h^{(1)}(u) = -h^{-2}k^{(1)}(u/h)$. The latter estimator is a bandit version of DPG (Silver et al., 2014). Like their OPE counterparts, these estimators suffer from high dependence on the nuisance estimates and potentially slow rates.

**Doubly Robust DPG** By differentiating our OPE estimator $\hat{J}^{\mathcal{K}}$ wrt $\theta$ we propose a new policy gradient estimator: $\hat{Z}^{\mathcal{K}} = \mathbb{P}_n[\psi^{\mathcal{K}}(\hat{q}, \hat{\pi}^b)]$ (recall $\hat{q}, \hat{\pi}^b$ are implicitly cross-fit in this notation), where

$$\psi^{\mathcal{K}}(q, \pi^b) = \left\{ \frac{K_h^{(1)}(a - \tau_\theta(s))\{r - q(s,a)\}}{\pi^b(a|s)} + \int K_h^{(1)}(a - \tau_\theta(s))q(s, a)\mathrm{d}a \right\} \nabla_\theta\tau_\theta(s). \quad (6)$$

Notice this does not involve explicit differentiation of $q$. Instead the convolution with $K_h^{(1)}$ essentially acts as an estimator for the derivative and may be computationally more stable than differentiating $\hat{q}$.

Similarly, by differentiating our OPE estimator $\hat{J}^{\mathcal{D}}$ wrt $\theta$, we propose $\hat{Z}^{\mathcal{D}} = \mathbb{P}_n[\psi^{\mathcal{D}}(\hat{q}, \hat{\pi}^b)]$, where

$$\psi^{\mathcal{D}}(q, \pi^b) = \left\{ \frac{K_h^{(1)}(a - \tau_\theta(s))\{r - q(s,\tau_\theta(s))\}}{\pi^b(\tau_\theta(s)|s)} + q^{(1)}(s, \tau_\theta(s)) \right\} \nabla_\theta\tau_\theta(s). \quad (7)$$

This can be understood as DPG *plus* a control variate. As in DPG, $\hat{Z}^{\mathcal{D}}$ requires differentiation of $\hat{q}$.

**Theorem 2.** *Suppose for $i = 1, 2$, $\mathbb{E}[\|\hat{\pi}^{b,[i]} - \pi^b\|_{2,\infty}] = o(1)$, $\mathbb{E}[\|\hat{q}^{[i]} - q\|_{2,\infty}] = o(1)$, $\mathbb{E}[\|\hat{\pi}^{b,[i]} - \pi^b\|_{1,\infty}\|\hat{q}^{[i]} - q\|_{1,\infty}] = o(n^{-1/2}h^{-3/2})$, $nh^7 = \mathcal{O}(1)$, $nh \to \infty$, $q(s, a), \pi^b(a \mid s), \hat{q}^{[i]}(s, a), \hat{\pi}^{b,[i]}(a \mid s)$ are thrice continuously differentiable functions wrt $a$, and $\hat{q}^{[i]}, \hat{\pi}^{b,[i]}$ are uniformly bounded by a constant. Then, the bias and variance of $\hat{Z}^{\mathcal{K}}$ are $\mathbb{E}[\hat{Z}^{\mathcal{K}}] - Z = 0.5h^2 M_2(k)\tilde{B} + o(n^{-1/2}h^{-3/2})$, $\mathrm{var}[\hat{Z}^{\mathcal{K}}] = \frac{\Omega_2^{(1)}(k)}{nh^3}\{\tilde{V} + o(1)\}$, where*

$$\tilde{B} = \mathbb{E}\left[\nabla\tau_\theta(S)q^{(3)}(S, \tau_\theta(S))\right], \quad \tilde{V} = \mathbb{E}\left[\otimes\nabla\tau_\theta(S)\frac{\mathrm{var}[R|S,\tau_\theta(S)]}{\pi^b(\tau_\theta(S)|S)}\right],$$

*where $\otimes v = vv^\top$. Setting $h = \Theta(n^{-1/7})$ yields the minimal MSE of order $\mathcal{O}(n^{-4/7})$.*

**Remark 3.** We can obtain a similar result for $\hat{Z}^{\mathcal{D}}$ with slightly different differentiability conditions; we omit the details. The best-achievable MSE rate in Theorem 2, $\mathcal{O}(n^{-4/7})$, matches the minimax rate for density gradient estimation in a Sobolev space of smoothness parameter 3 (Korostelev, 2011). We therefore conjecture the rate for $Z$ is minimax optimal under the assumptions of the theorem.

**Remark 4.** As in the case of OPE, there are two crucial advantages of $\hat{Z}^{\mathcal{K}}, \hat{Z}^{\mathcal{D}}$ over ISPG and DPG estimators in Eq. (5). First, the required convergence rates on nuisances are weaker and we do not depend on the particular estimators. On the other hand, ISPG and DPG do not have convergence guarantees given only rate conditions on $\hat{\pi}^b$ and $\hat{q}$, respectively. Second, our leading constant in the variance is smaller than ISDP with an oracle behavior policy, just as in Corollary 1.

**Remark 5** ($\hat{Z}^{\mathcal{K}}$ vs $\hat{Z}^{\mathcal{D}}$). Unlike $\hat{Z}^{\mathcal{K}}$, the estimator $\hat{Z}^{\mathcal{D}}$ does not involve integration, which may be computationally preferable. However, direct differentiation of $q$-functions can often be statistically unstable (see also Athey and Wager, 2017, Section 5.2). In our empirical results in Section 5, we indeed find $\hat{Z}^{\mathcal{K}}$ is superior. Moreover, when we use the Gaussian kernel and polynomial sieve nuisance estimators, the integration can be easily done analytically.

# 4 Offline RL with Deterministic Policies

We next discuss how to extend the ideas from the previous section to the RL setting where $H \geq 1$ in general. In this setting there are actually different ways to account for the IS part of the estimator, leading to different dependence on horizon. Throughout this section, we will assume the densities $p_t(r_t \mid s_t, a_t), p_{t+1}(s_{t+1} \mid s_t, a_t)$ are thrice continuously differentiable wrt action and $\| \int |p_t^{(i)}(r_t \mid s_t, a_t)| dr_t \|_\infty \leq G_1^{(i)} < \infty$, $\| \int |p_t^{(i)}(s_t \mid s_{t-1}, a_{t-1})| ds_t \|_\infty \leq G_2^{(i)} < \infty$ for $i = 1, 2, 3$. We also assume all of nuisance estimators introduced in this section are uniformly bounded by some constant.

## 4.1 Off-Policy Deterministic Policy Evaluation

Motivated by DR OPE using cumulative density ratios for the case of stochastic policies (Jiang and Li, 2016), we propose analogous extensions of $\hat{J}^{\mathcal{K}}, \hat{J}^{\mathcal{D}}$ for $H \geq 1$: Cumulative DR case $\mathcal{K}$ (CDRK) $\hat{J}_{\mathrm{cdr}}^{\mathcal{K}} = \mathbb{P}_n[\phi_{\mathrm{cdr}}^{\mathcal{K}}(\hat{q}_t^{\mathcal{K}}, \hat{\pi}_t^b)]$ and Cumulative DR case $\mathcal{D}$ (CDRD) $\hat{J}_{\mathrm{cdr}}^{\mathcal{D}} = \mathbb{P}_n[\phi_{\mathrm{cdr}}^{\mathcal{D}}(\hat{q}_t^{\pi^e}, \hat{\pi}_t^b)]$, where

$$\phi_{\mathrm{cdr}}^{\mathcal{K}}(q_t^{\mathcal{K}}, \pi_t^b) = v_1^{\mathcal{K}} + \sum_{t=1}^H \lambda_t^{\mathcal{K}}\{r_t - q_t^{\mathcal{K}}(s_t, a_t) + v_{t+1}^{\mathcal{K}}\}, \quad \lambda_t^{\mathcal{K}} = \prod_{k=1}^t \frac{K_h(a_k - \tau_k(s_k))}{\pi_k^b(a_k|s_k)},$$

$$\phi_{\mathrm{cdr}}^{\mathcal{D}}(q_t^{\pi^e}, \pi_t^b) = v_1^{\pi^e} + \sum_{t=1}^H \lambda_t^{\pi^e}\{r_t - q_t^{\pi^e}(s_t, \tau_t(s_t)) + v_{t+1}^{\pi^e}\}, \quad \lambda_t^{\pi^e} = \prod_{k=1}^t \frac{K_h(a_k - \tau_k(s_k))}{\pi_k^b(\tau_k(s_k)|s_k)},$$

and where $q_t^{\mathcal{K}}$ is the $q$-function associated with the kernelized evaluation policy, $\pi_t^{e,\mathcal{K}}(a_t \mid s_t) = K_h(a_t - \tau_\theta(s_t))$. We discuss the estimation of nuisances in Remark 7. Recall we use cross-fitting.

**Theorem 3.** *Suppose for* $j \leq H, i = 1, 2, \mathbb{E}[\|\hat{\pi}_j^{b,[i]} - \pi_j^b\|_{1,\infty}] = o(1), \mathbb{E}[\|\hat{q}_j^{\mathcal{K},[i]} - q_j^{\mathcal{K}}\|_{1,\infty}] = o(1),$ $\mathbb{E}[\|\hat{\pi}_j^{b,[i]} - \pi_j^b\|_\infty \|\hat{q}_j^{\mathcal{K},[i]} - q_j^{\mathcal{K}}\|_\infty] = o(n^{-1/2}h^{-H/2}), nh^{H+4} = \mathcal{O}(1), nh \to \infty.$ *Then, we have* $\mathbb{E}[\hat{J}_{\mathrm{cdr}}^{\mathcal{K}}] - J = 0.5h^2 M_2(k) B_H^{\mathrm{cdr}} + o(n^{-1/2}h^{-H/2}), \mathrm{var}[\hat{J}_{\mathrm{cdr}}^{\mathcal{K}}] = \frac{\Omega_2^H(k)}{nh^H}\{V_H^{\mathrm{cdr}} + o(1)\},$ *where*

$$B_H^{\mathrm{cdr}} = \sum_{t=1}^H \mathbb{E}_{p_\pi^e}\left[\frac{r_t p_t^{(2)}(r_t|s_t, \tau_t(s_t))}{p_t(r_t|s_t, \tau_t(s_t))}\right] + \sum_{j=1}^{t-1} \mathbb{E}_{p_\pi^e}\left[\frac{r_t p_{j+1}^{(2)}(s_{j+1}|s_j, \tau_j(s_j))}{p_{j+1}(s_{j+1}|s_j, \tau_j(s_j))}\right],$$

$$V_H^{\mathrm{cdr}} = \mathbb{E}_{p_\pi^e}\left[\frac{1}{\prod_{i=1}^H \pi_i^b(\tau_i(s_i)|s_i)}\mathrm{var}[r_H \mid s_H, a_H]\right].$$

*In the above, setting* $h = \Theta(n^{-1/(H+4)})$ *yields the minimal MSE of order* $\mathcal{O}(n^{-4/(H+4)})$.

**Remark 6** (Curse of Horizon in Rate). Notice CDRK and CDRD have convergence *rate* that deteriorates as the horizon grows. In usual OPE for stochastic policies, Liu et al. (2018); Kallus and Uehara (2019a, 2020a) show that using cumulative IS leads to MSE with leading constant that grows exponentially in horizon, but it still has rate $\mathcal{O}(1/n)$. Thus, the curse of horizon is even more detrimental for deterministic policies. However, CDRK and CDRD technically work also for *non*-Markov decision processes. Next, we will tackle the curse in rate by leveraging Markovian structure, following Kallus and Uehara (2020a).

Motivated by DR OPE using marginal density ratios for stochastic policies (Kallus and Uehara, 2020a), we propose Marginal DR case $\mathcal{K}$ (MDRK) for deterministic OPE: $\hat{J}_{\mathrm{mdr}}^{\mathcal{K}} = \mathbb{P}_n[\phi_{\mathrm{mdr}}^{\mathcal{K}}(\hat{q}_t^{\mathcal{K}}, \hat{w}_t^{\mathcal{K}}, \hat{\pi}_t^b)]$, where

$$\phi_{\mathrm{mdr}}^{\mathcal{K}}(q_t^{\mathcal{K}}, w_t^{\mathcal{K}}, \pi_t^b) = v_1^{\mathcal{K}}(s_1) + \sum_{t=1}^H w_t^{\mathcal{K}}(s_t)\pi_t^b(a_t \mid s_t)K_h(a_t - \tau_t(s_t))(r_t - q_t^{\mathcal{K}}(s_t, a_t) + v_{t+1}^{\mathcal{K}}(s_{t+1}))$$

and $w_t^{\mathcal{K}}(s_t) = p_{\pi^e,\mathcal{K}}(s_t)/p_{\pi^b}(s_t)$ is the marginal density ratio associated with $\pi_t^{e,\mathcal{K}}$. Again, a similar estimator, Marginal DR case $\mathcal{D}$ (MDRD), $\hat{J}_{\mathrm{mdr}}^{\mathcal{D}}$, is constructed by replacing $q_t^{\mathcal{K}}(s_t, a_t), w_t^{\mathcal{K}}(s_t), \pi_t^b(a_t \mid s_t)$ with $q_t^{\pi^e}(s_t, \tau_t(s_t)), w_t^{\pi^e}(s_t), \pi_t^b(\tau_t(s_t) \mid s_t)$.

**Theorem 4.** *Suppose for* $j \leq H, i = 1, 2, \mathbb{E}[\|\hat{\pi}_j^{b,[i]} - \pi_j^b\|_{1,\infty}] = o(1), \mathbb{E}[\|\hat{w}_j^{\mathcal{K},[i]} - w_j^{\mathcal{K}}\|_\infty] = o(1), \mathbb{E}[\|\hat{q}_j^{\mathcal{K},[i]} - q_j^{\mathcal{K}}\|_{1,\infty}] = o(1), \mathbb{E}[\max\{\|\hat{\pi}_j^{b,[i]} - \pi_j^b\|_\infty, \|\hat{w}_j^{\mathcal{K},[i]} - w_j^{\mathcal{K}}\|_\infty\}\|\hat{q}_j^{\mathcal{K},[i]} - q_j^{\mathcal{K}}\|_\infty] =$

$o(n^{-1/2}h^{-1/2})$, $nh^5 = \mathcal{O}(1)$, $nh \to \infty$. Then, the bias of $\hat{J}^{\mathcal{K}}_{\mathrm{mdr}}$ is the same as $\hat{J}^{\mathcal{K}}_{\mathrm{cdr}}$ in Theorem 3 and its variance is $\mathrm{var}[\hat{J}^{\mathcal{K}}_{\mathrm{mdr}}] = \frac{\Omega_2(k)}{nh} V^{\mathrm{mdr}}_H + o(n^{-1}h^{-1})$, where

$$V^{\mathrm{mdr}}_H = \sum_{t=1}^{H} \mathbb{E}_{p_\pi^e}\left[ \frac{w_t^{\pi^e}(s_t)}{\pi_t^b(\tau_t(s_t)|s_t)} \mathrm{var}[r_t + v_{t+1}^{\pi^e}(s_{t+1}) \mid s_t, \tau_t(s_t)] \right].$$

*Setting $h = \Theta(n^{-1/5})$ yields the minimal MSE of order $\mathcal{O}(n^{-4/5})$. Specifically, if $h = cn^{-1/5}$, then*

$$\mathbb{E}[(\hat{J}^{\mathcal{K}}_{\mathrm{mdr}} - J)^2] \le n^{-4/5} R_{\max}^2 H^2 \left\{ \frac{c^4 M_2^2(k)}{4}(G_1^{(2)} + \frac{(H-1)}{2}G_2^{(2)})^2 + \frac{C_1 C_2 \Omega_2(k)}{c} \right\} + o(n^{-4/5}).$$

Notice the minimal MSE rate is the same as in the bandit case. We therefore conjecture the rate to be minimax optimal. More crucially, it does not suffer from the curse of horizon in rate. Moreover, the dependence of the leading constant is polynomial in horizon, $\mathcal{O}(H^4)$. The leading constant is smaller when $C_1, C_2, G_1^{(2)}, G_2^{(2)}$ are smaller, i.e., when the behavior policy is closer to the evaluation policy and the reward and transition densities are smoother.

## 4.2 Off-Policy Deterministic Policy Gradient Estimation

We next construct deterministic policy gradient estimators for RL. By differentiating $\phi^{\mathcal{K}}_{\mathrm{mdr}}$, we obtain the Marginal PG case $\mathcal{K}$ (MPGK) estimator, $\hat{Z}^{\mathcal{K}}_{\mathrm{mpg}} = \mathbb{P}_n[\psi^{\mathcal{K}}_{\mathrm{mpg}}(\hat{q}_t^{\mathcal{K}}, \hat{w}_t^{\mathcal{K}}, \hat{d}_t^{q^{\mathcal{K}}}, \hat{d}_t^{w^{\mathcal{K}}}, \hat{\pi}_t^b)]$, where

$$\psi^{\mathcal{K}}_{\mathrm{mpg}}(q_t^{\mathcal{K}}, w_t^{\mathcal{K}}, d_t^{q^{\mathcal{K}}}, d_t^{w^{\mathcal{K}}}, \pi_t^b) = d_1^{v^{\mathcal{K}}} + \sum_{t=1}^{H} \frac{d_t^{w^{\mathcal{K}}} K_h(a_t - \tau_t(s_t))}{\pi_t^b(a_t|s_t)}(r_t - q_t^{\mathcal{K}} + v_{t+1}^{\mathcal{K}})$$

$$+ \sum_{t=1}^{H} \frac{w_t^{\mathcal{K}}}{\pi_t^b(a_t|s_t)}\left( K_h^{(1)}(a_t - \tau_t(s_t))(r_t - q_t^{\mathcal{K}} + v_{t+1}^{\mathcal{K}})\nabla\tau_t(s_t) + K_h(a_t - \tau_t(s_t))(-d_t^{q^{\mathcal{K}}} + d_{t+1}^{v^{\mathcal{K}}}) \right),$$

$q_t^{\mathcal{K}} = q_t^{\mathcal{K}}(s_t, a_t), d_t^{q^{\mathcal{K}}}(s_t, a_t) = \nabla q_t^{\mathcal{K}}(s_t, a_t), d_t^{v^{\mathcal{K}}}(s_t) = \nabla v_t^{\mathcal{K}}(s_t), d_t^{w^{\mathcal{K}}}(s_t) = \nabla w_t^{\mathcal{K}}(s_t),$

$v_t^{\mathcal{K}}(s_t) = \int q_t^{\mathcal{K}}(s_t, a_t) K_h(a_t - \tau_t(s_t))\, \mathrm{d}a_t,$

$d_t^{v^{\mathcal{K}}}(s_t) = \int \left\{ d_t^{q^{\mathcal{K}}}(s_t, a_t) K_h(a_t - \tau_t(s_t)) + q_t(s_t, a_t) K_h^{(1)}(a_t - \tau_t(s_t)) \nabla\tau_t(s_t) \right\} \mathrm{d}a_t.$

Notice we only estimate the nuisances $q_t^{\mathcal{K}}, w_t^{\mathcal{K}}, d_t^{q^{\mathcal{K}}}, d_t^{w^{\mathcal{K}}}, \pi_t^b$; then estimates for $v_t^{\mathcal{K}}, d_t^{v^{\mathcal{K}}}$ are defined in terms of these. The Marginal PG case $\mathcal{D}$ (MPGD) estimator, $\hat{Z}^{\mathcal{D}}_{\mathrm{mpg}}$, is similarly defined by replacing the nuisances in $\hat{Z}^{\mathcal{K}}_{\mathrm{mpg}}$ by $q_t^{\pi^e}(s_t, \tau_t(s_t)), w_t^{\pi^e}(s_t), d_t^{q^{\pi^e}}(s_t, \tau_t(s_t)), d_t^{w^{\pi^e}}(s_t), \pi_t^b(\tau_t(s_t) \mid s_t)$. Again, note $\hat{Z}^{\mathcal{D}}_{\mathrm{mpg}}$ involves a differentiation while $\hat{Z}^{\mathcal{K}}_{\mathrm{mpg}}$ does not but involves an integration, as in Remark 5.

**Theorem 5.** *Suppose for $j \le H, i = 1, 2$, $\mathbb{E}[\|\hat{w}_j^{\mathcal{K},[i]} - w_j^{\mathcal{K}}\|_\infty] = o(1)$, $\mathbb{E}[\|\hat{\pi}^{b,[i]}_j - \pi^b\|_{2,\infty}] = o(1)$, $\mathbb{E}[\|\hat{d}_j^{w^{\mathcal{K}},[i]} - d_j^{w^{\mathcal{K}}}\|_\infty] = o(1)$, $\mathbb{E}[\|\hat{q}_j^{\mathcal{K},[i]} - q_j^{\mathcal{K}}\|_{2,\infty}] = o(1)$, $\mathbb{E}[\|\hat{d}_j^{q^{\mathcal{K}},[i]} - d_j^{q^{\mathcal{K}}}\|_{1,\infty}] = o(1)$, $\mathbb{E}[\max\{\|\hat{w}_j^{\mathcal{K},[i]} - w_j^{\mathcal{K}}\|_\infty, \|\hat{\pi}_j^{b,[i]} - \pi_j^b\|_{1,\infty}, \|\hat{d}_j^{w^{\mathcal{K}},[i]} - d_j^{w^{\mathcal{K}}}\|_\infty\} \max\{\|\hat{q}_j^{\mathcal{K},[i]} - q_j^{\mathcal{K}}\|_{1,\infty}, \|\hat{d}_j^{q^{\mathcal{K}},[i]} - d_j^{q^{\mathcal{K}}}\|_\infty\}] = o(n^{-1/2}h^{-3/2})$, $nh^7 = \mathcal{O}(1)$, $nh \to \infty$. Then, we have $\mathbb{E}[\hat{Z}^{\mathcal{K}}_{\mathrm{mpg}}] - Z = 0.5h^2 M_2(k)\tilde{B}^{\mathrm{mpg}}_H + o(n^{-1/2}h^{-3/2})$, $\mathrm{var}[\hat{Z}^{\mathcal{K}}_{\mathrm{mpg}}] = \frac{\Omega_2^{(1)}(k)}{nh^3}\tilde{V}^{\mathrm{mpg}}_H + o(n^{-1}h^{-3})$, where*

$$\tilde{B}^{\mathrm{mpg}}_H = \sum_{t=1}^{H}\left( \nabla\mathbb{E}_{p_\pi^e}\left[ \frac{r_t p_t^{(2)}(r_t|s_t, \tau_t(s_t))}{p_t(r_t|s_t, \tau_t(s_t))} \right] + \sum_{l=1}^{t-1} \nabla\mathbb{E}_{p_\pi^e}\left[ \frac{r_t p_{l+1}^{(2)}(s_{l+1}|s_l, \tau_l(s_l))}{p_{l+1}(s_{l+1}|s_l, \tau_l(s_l))} \right] \right),$$

$$\tilde{V}^{\mathrm{mpg}}_H = \sum_{t=1}^{H} \mathbb{E}_{p_\pi^e}\left[ \frac{w_t^{\pi^e}(s_t)}{\pi_t^b(\tau_t(s_t)|s_t)} \mathrm{var}_{p_\pi^e}[r_t + q_{t+1}^{\pi^e}(s_{t+1}, \tau_{t+1}(s_{t+1})) \mid s_t, \tau_t(s_t)] \otimes \nabla\tau_t(s_t) \right].$$

*Setting $h = \Theta(n^{-1/7})$ yields the minimal MSE of order $\mathcal{O}(n^{-4/7})$. Specifically, if $h = cn^{-1/7}$, the operator norm of the MSE is bounded by*

$$\frac{R_{\max}^2 H^2 \Upsilon^2}{n^{4/7}}\left( \frac{c^4 M_2^2(k)}{4}\left\{ G_1^{(3)} + \frac{(H-1)}{2}\{G_1^{(2)}G_2^{(1)} + G_1^{(1)}G_2^{(2)} + G_2^{(3)}\} + \frac{(H-1)(H-2)}{3}G_2^{(1)}G_2^{(2)} \right\}^2 + \frac{C_1 C_2 \Omega_2^{(1)}(k)}{c^3} \right).$$

Note again that the MSE rate $\mathcal{O}(n^{-4/7})$ is slower than the usual $\mathcal{O}(1/n)$ efficient rate for off-policy gradient estimation with stochastic policies (Kallus and Uehara, 2020b). Nonetheless, it matches the bandit case and we therefore conjecture it is minimax optimal. More importantly, we note that it alleviates the curse of horizon, both in rate and in leading constant. We can also derive corresponding CPGK and CPGD estimators by differentiating the CDRK and CDRD estimating functions, but these *will* suffer from the curse of horizon in rate, as in Theorem 3.

**Remark 7** (Estimation of nuisance functions). Our OPE and off-policy policy gradient estimators depend on estimating some nuisances. A unique and new feature of our estimators and analysis compared to previous deterministic off-policy estimators is that the MSE guarantees do not depend on the particular nuisance estimator used and we make no assumptions except for their (slow) convergence rate. The estimation of $q_t, w_t$ for stochastic policies is discussed in Kallus and Uehara (2020a) and of $d_t^q, d_t^w$ in Kallus and Uehara (2020b). These can be applied directly to estimate $q_t^{\mathcal{K}}, w_t^{\mathcal{K}}, d_t^{q^{\mathcal{K}}}, d_t^{w^{\mathcal{K}}}$ since the kernelized evaluation policy, $\pi^{e,\mathcal{K}}$, is stochastic. The estimation of $q_t^{\pi^e}$ is the same for deterministic policies and a small adjustment can also be made for $d_t^{q^{\pi^e}}$ as we explain in Appendix B. The estimation $w_t^{\pi^e}, d_t^{w^{\pi^e}}$ for deterministic policies is difficult, but we can simply use estimates of $w_t^{\mathcal{K}}, d_t^{w^{\mathcal{K}}}$ as estimates for $w_t^{\pi^e}, d_t^{w^{\pi^e}}$, which is essentially a kernel density estimation approach for the densities in the latter. For additional detail, refer to Appendix B.

**Remark 8** (Policy learning algorithms). To do offline RL to learn a deterministic policy, we can combine any type of gradient-based optimization algorithm with our estimated gradients. A simple gradient ascent is given as an example in Appendix C and used in the experiments in the next section. Following Kallus and Uehara (2020b) we can also combine standard results for gradient ascent with our error bounds to obtain a regret guarantee. Since the proof is exactly the same, simply plugging in our error bounds instead, we omit the details and refer the reader to Kallus and Uehara (2020b).

# 5 Experiments

We next conduct an experiment in a very simple environment to confirm the theoretical guarantees of the proposed estimators. More extensive experimentation remains future work. The setting is as follows. Set $\mathcal{S}_t = \mathbb{R}$, $\mathcal{A}_t = \mathbb{R}$, $s_0 = 0$. Then, set the transition dynamics as $s_t = a_{t-1} - s_{t-1} + \mathcal{N}(0, 0.3^2)$, the reward as $r_t = -s_t^2$, the behavior policy as $\pi^{\mathrm{b}}(a \mid s) = \mathcal{N}(0.8s, 1.0^2)$, the deterministic evaluation policy as $\tau_t(s_t) = \theta s_t$, and the horizon as $H = 20$. Note that in this setting, the optimal policy is given by $\theta^* = 1$. We compare CPGK, CPGD, MPGK, MPGD using the Gaussian kernel with PG. The nuisance functions $q$, $w$, $d^q$, $d^w$ (and their case $\mathcal{K}$ equivalents) are estimated using polynomial sieve regressions (Chen, 2007). We assume the behavior policy is known. Since $q$ is estimated by polynomials and $k$ is Gaussian, we can compute the integrals in MPGK and CPGK analytically. We use the same estimated $q$ in PG. We choose $h$ by bootstrapping the estimator for each of $h \in \{0.05, 0.1, 0.25, 0.5\}$ and choosing that with smallest bootstrap variance.

First, in Fig. 1, we compare the MSE of gradient estimators at $\theta = 1.0$ over 100 replications for each of $n = 200, 400, 600, 800$. We find that the performance of MPGK is far superior to all other estimators in terms of MSE, which confirms our theoretical results. Interestingly, the performance of MPGD is slightly worse than CPGD. The possible reason is it is more difficult to estimate $w$ than $w^{\mathcal{K}}$. The reasonably good performance of CDGD and CDGK can be attributed to the known $\lambda_t^{\mathcal{D}}, \lambda_t^{\mathcal{K}}$, which ensures less sensitivity to the $q$-estimation due to the doubly robust error structure.

Second, in Fig. 2, we apply gradient ascent (see Appendix C) with $\alpha_t = 0.05$, $T = 50$, and $\hat{\theta}_1$ randomly chosen from $[0.8, 1.2]$. We only run the bootstrap for $\hat{\theta}_1$ and then keep the same $h$ for the next iterations. We compare the regret of the final policy for the different policy gradient estimators, i.e., $J(\theta^*) - J(\hat{\theta}_{50})$, averaging over 100 replications of the experiment for each of $n = 200, 400, 600, 800$. Again, the performance of MPGK is superior to other estimators also in terms of regret.

# 6 Conclusion and Future work

We developed doubly robust versions of DPG and showed that they can circumvent issues of curse of horizon and of dependence on nuisances such as $q$-estimates. Theoretically, a next question may be showing the rates we obtain are minimax optimal by appealing to minimax theory for nonparametric density estimation (Korostelev, 2011). A more practical next step may be to apply this in larger-scale RL environments. Offline RL in large-scale environments is notoriously difficult (Fujimoto et al., 2019). We therefore expect it necessary to combine several heuristics, such as gradient updates to nuisance estimates and adaptive step sizes, to make the algorithm work well in practice.

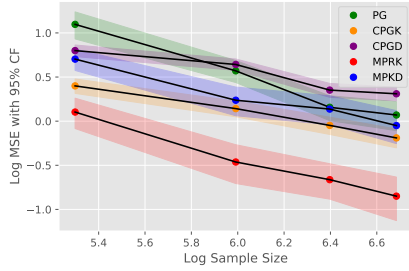 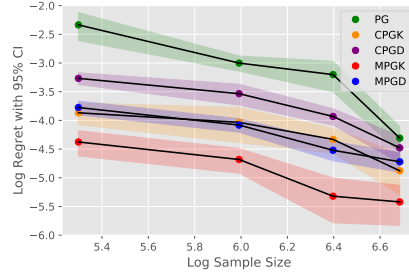

Fig. 1: MSE of gradient estimation with 95% CF     Fig. 2: Regret after gradient ascent with 95% CF

## Acknowledgement

This material is based upon work supported by the National Science Foundation under Grant No. 1846210

Masatoshi Uehara was supported in part by MASASON Foundation.

## Broader Impact

As a contribution to offline RL, our work is of particular importance for RL in the context of social and medical sciences, where experimentation is limited and observational data must be used. Deterministic policies are particularly important for real applications of offline RL because optimal policies are deterministic (up to ties). There exist various examples of the use of deterministic policies in epidemiology (Wu et al., 2018), political science (Imai and van Dyk, 2004), and economics (Colangelo and Lee, 2019). We provide some of the first theoretical results on convergence rates in sequential RL settings. Our work contributes both to the theoretical understanding of deterministic policies in RL and to practical methods for evaluating and learning these.

That said, it is well understood that there is generally a gap between theory and practice in RL as it is applied to very complex and large-scale systems, and valid convergence rate guarantees need not directly translate by themselves into practical success in real, large-scale settings. It is therefore important to keep in mind practical heuristics, stopgaps, and approximations from applied RL when translating this work into practice. The systematic investigation of the use of these in the context of deterministic-policy offline RL may require additional future work.

There are also several potential dangers to be cognizant of when applying offline RL tools in general. First, the presence of large unobserved confounding in the data (in our setting this would manifest as large violations of the MDP model) can bias evaluation and, unchecked, may potentially hide harms done by the policy being evaluated or the policy learned by off-policy optimization. Second, if the observational data is not representative of the population, that is, there is a covariate shift in the initial state distribution, then the evaluation both of value and of gradient will reflect these biases and correspondingly be unrepresentative, under-emphasizing value to some parts of the population and over-emphasizing value to others, leading to learned policies that possibly benefit over-represented subpopulations more than others. More generally, even without covariate shift, here we focused on evaluation and optimization of *average* welfare, which may average the harms to some and the benefits to others; it may therefore be important in some applications to also conduct auxiliary evaluations on certain protected subgroups to ensure equal impact, which can be done by segmenting the data.

## Footnotes

[2]In the special case when $q$ (or $\pi^b$, respectively) is estimated using kernel estimators, we can obtain an MSE rate $\mathcal{O}(n^{-4/5})$ for DM (or IS, respectively) but it requires much stricter conditions on the smoothness, including smoothness in the state variable in addition to smoothness in action (Hsu et al., 2018; Lee, 2018).

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
