[Supplementary Material · full_draft.pdf]

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

# A    Notation

Table 2: Notation

| | |
|---|---|
| $p_1(s)$ | Initial distributions |
| $p_\pi(\cdot)$ | Induced distribution by a MDP and a policy $\pi$ |
| | $p_\pi(j_{r_{H+1}}) = p_1(s_1)\prod_{t=1}^{H}\pi(a_t \mid s_t)p(r_t \mid s_t, a_t)p(s_{t+1} \mid s_t, a_t).$ |
| $p_\tau(\cdot)$ | $p_\pi(j_{r_{H+1}}) = p_1(s_1)\prod_{t=1}^{H} p(r_t \mid s_t, \tau_t(s_t))p(s_{t+1} \mid s_t, \tau_t(s_t))$ |
| $p_\mathcal{K}(\cdot)$ | $p_\pi(j_{r_{H+1}}) = p_1(s_1)\prod_{t=1}^{H} K_h(a_t - \tau_t(s_t))p(r_t \mid s_t, a_t)p(s_{t+1} \mid s_t, a_t)$ |
| $H$ | Horizon |
| $h$ | Bandwidth |
| $\mathcal{T}^{\langle i\rangle}$ | i-th data |
| $\pi^{\mathrm{b}}, \pi^{\mathrm{e}}, \pi^{e,\mathcal{K}}$ | Behavior policy, Evaluation policy, Kernelized policy |
| $J(\theta), Z(\theta)$ | Value, Gradient |
| $k(\cdot), K_h(x)$ | Kernel, Normalized kernel $h^{-1}k(h^{-1}x)$ |
| $\tau, \tau_\Theta$ | Deterministic policy with a parameter $\theta \in \Theta$ |
| $\mathbb{E}[\cdot]$ | Expectation wrt random variable generated by MDP and a behavior policy |
| $\mathbb{E}_{\pi^{\mathrm{e}}}[\cdot]$ | Expectation wrt random variable generated by a MDP and a policy $\pi^{\mathrm{e}}$ |
| $\mathbb{E}_{\pi^{\mathrm{e}}}[\tau]$ | Expectation wrt random variable generated by a MDP and a policy $\delta(a = \tau_t(s))$ |
| $\mathcal{H}_{S_t}, \mathcal{H}_{A_t}$ | History $(S_0, A_0, S_1, A_1, \cdots, S_t), (S_0, A_0, S_1, A_1, \cdots, A_t)$ |
| $h^u_{s_t}$ | History $(s_0, u_0, s_1, u_1, \cdots, s_t)$ |
| $M_2(k)$ | Second moment of kernel, $\int u^2 k(u)\mathrm{d}u$ |
| $\Omega^{(i)}(k)$ | Roughness of kernel, $\int k^{2(i)}(u)\mathrm{d}u$ |
| $v^{\mathcal{K}}, q^{\mathcal{K}}$ | Value, Q-function wrt a kernelized policy and a MDP |
| $v^{\pi^{\mathrm{e}}}, q^{\pi^{\mathrm{e}}}$ | Value, Q-function wrt a deterministic policy and a MDP |
| $q^{(i)}(s, a)$ | i-th Differentiation of $q(s,a)$ wrt actions |
| $w_t^{\mathcal{K}}(s_t), w_t^{\pi^{\mathrm{e}}}(s_t)$ | $p_\mathcal{K}(s_t)/p_{\pi^{\mathrm{b}}}(s_t), p_\tau(s_t)/p_{\pi^{\mathrm{b}}}(s_t)$ |
| $\Upsilon$ | $\|\| \mid \otimes \tau_t(S)\|_\infty\|_{\mathrm{op}} \le \Upsilon$ |
| $R_{\max}$ | $\|R_t\|_\infty \le R_{\max}$ |
| $C_1$ | $\|1/\pi^{\mathrm{b}}(a\mid s)\| \le C_1$ |
| $C_2$ | $\|w_t^{\pi^{\mathrm{e}}}(s)\| \le C_2$ |
| $\nabla$ | Differentiation wrt $\theta$ |
| $\|\cdot\|_2$ | $L^2$-norm $\{\mathbb{E}[f^2]\}^{1/2}$ |
| $\|\cdot\|_{\mathrm{op}}$ | Operator norm |
| $\|\cdot\|_{i,\infty}$ | $\max_{j=0,\cdots,i}\|\cdot^{(j)}\|_\infty$ |
| $d_t^{w^{\mathcal{K}}}(s_t), d_t^{w^{\pi^{\mathrm{e}}}}(s_t)$ | $\nabla w_t^{\mathcal{K}}(s_t), \nabla w_t^{\pi^{\mathrm{e}}}(s_t)$ |
| $d_t^{q^{\mathcal{K}}}(s_t, a_t), d_t^{q^{\pi^{\mathrm{e}}}}(s_t, a_t)$ | $\nabla q_t^{\pi^{\mathrm{e}}}(s_t, a_t)$ |
| $G_1^{(i)}$ | $\|\int p^{(i))}(r\mid s,a)\mathrm{d}r\|_\infty \le G_1^{(i)}$ |
| $G_2^{(i)}$ | $\|\int p^{(i))}(s'\mid s,a)\mathrm{d}s'\|_\infty \le G_2^{(i)}$ |
| $\otimes a$ | $aa^\top$ |
| $f(n) = \mathcal{O}(n^a)$ | $f(n)$ is bounded above by $n^a$ asymptotically |
| $f(n) = \Theta(n^a)$ | $f(n)$ is bounded both above and below by $n^a$ asymptotically |
| $f(n) = \mathcal{O}_p(n^a)$ | $f(n)/n^a$ is bounded in probability |
| $f(n) = \circ_p(n^a)$ | $f(n)/n^a$ converges to $0$ in probability |
| $A_n \lesssim B_n$ | $\exists C, A_n < CB_n, C$ is a universal problem independent constant |

# B  Nuisance Estimations

Our algorithm allows any estimators for $q$-functions and marginal ratios to be used. In this section we discuss some standard ways to estimate these nuisance functions.

## B.1  Estimation of $q$-functions and their $\theta$-gradients

In the tabular case, a model-based approach is the most common way to estimate $q$-functions of $\pi^{\mathrm{e}}$ from off-policy data. In the non-tabular case, we have to rely on some function approximation. The key equation to derive these methods is the Bellman equation:

$$q_t(s_t, a_t) = \mathbb{E}[r_t + q_{t+1}(s_{t+1}, \pi^{\mathrm{e}}) \mid s_t, a_t].$$

There is also an equivalent equation for $d^q$. If $\pi^{\mathrm{e}}$ is stochastic we may use

$$d_j^q(s_j, a_j) = \mathbb{E}[d_{j+1}^v(s_{j+1}) \mid s_j, a_j], \ d_j^v(s_j) = \mathbb{E}_{\pi^{\mathrm{e}}}[d_j^q + g_j q_j \mid s_j],$$

where $q_t(s_t, \pi) = \int q_t(s_t, a_t)\pi(a_t \mid s_t)\mathrm{d}a_t$ and $g_t = \log \pi_t^{\mathrm{e}}(a_t \mid s_t)$. When $\pi^{\mathrm{e}}$ is deterministic, we instead have

$$d_j^q(s_j, a_j) = \mathbb{E}[d_{j+1}^v(s_{j+1}) \mid s_j, a_j], \ d_j^v(s_j) = d_j^q(s_j, \tau_j(s_j)) + q_j^{(1)}(s_j, \tau_j(s_j))\nabla\tau_j(s_j).$$

One of the most common ways to operationalize this is using fitted $q$-iteration (Antos et al., 2008; Le et al., 2019:

- Set $\hat{q}_{H+1} \equiv 0$.
- For $t = H, \ldots, 1$:
  - Estimate $\hat{q}_t$ by regressing $r_t + \hat{q}_{t+1}(s_{t+1}, \pi^e)$ onto $s_t, a_t$.

Similarly, Kallus and Uehara (2020b) proposed an analogous estimation method for $d_j^q$:

- Set $\hat{d}_H^q = 0$.
- For $t = H, \ldots, 1$:
  - Estimate $\hat{d}_j^q$ by regressing $\hat{d}_{j+1}^v(s_{j+1})$ onto $s_j, a_j$.

The above approaches can be regarded as a dynamic programming approach. When $\pi^{\mathrm{e}}$ is stochastic, another approach is a Monte Carlo approach based on the equations:

$$q_j(s_j, a_j) = \mathbb{E}\left[\sum_{t=j}^{H} r_t | s_j, a_j\right], \quad d_j^q(s_j, a_j) = \mathbb{E}\left[\sum_{t=j+1}^{H} r_t \lambda_{j+1:t} \sum_{\ell=j+1}^{t} g_\ell \mid s_j, a_j\right].$$

Based on this, we can simply estimate $q$ by regressing $\sum_{t=j}^{H} r_t$ on $s_j, a_j$ and $d^q$ by regressing $\sum_{t=j+1}^{H} r_t \lambda_{j+1:t} \sum_{\ell=j+1}^{t} g_\ell$ on $a_j, s_j$.

## B.2  Estimation of Marginal Density Ratios

When $\mathcal{S}$ is finite, a model based approach (Yin and Wang, 2020) would be a competitive way to estimate marginal density ratios:

$$\hat{w}_t^{\pi^e}(s_t) = \frac{1}{\hat{p}_{\pi_t^{\mathrm{b}}}(s_t)} \int \hat{p}_t(s_t|s_{t-1}, \tau(s_{t-1})) \prod_{k=0}^{t-1} (\hat{p}_k(s_k|s_{k-1}, \tau(s_{k-1}))) \, \mathrm{d}(\mathcal{H}_{a_{t-1}}),$$

where $\hat{p}_t, \hat{p}_{\pi_t^{\mathrm{b}}}$ is each an empirical frequency (histogram) estimator. For general state space, we have to rely on some function approximation methods. When the target policy is stochastic, we have the following equations: $j \leq H$,

$$w_j(s_j) = \mathbb{E}[\lambda_{j-1}|s_j], \ d_j^w = \mathbb{E}\left[\lambda_{0:j-1} \sum_{\ell=0}^{j-1} g_\ell \mid s_j\right].$$

Thus, for example, $w_j$ is estimated by regressing $\lambda_{j-1}$ onto $s_j$, and $d_j^w$ is estimated by regressing $\lambda_{0:j-1} \sum_{\ell=0}^{j-1} g_\ell$ onto $s_j$ (Kallus and Uehara, 2020a,b). When the evaluation policy is deterministic, it is difficult to estimate directly.

## C  Off-policy Optimization

Our estimated policy gradients can be used in any gradient-based optimization algorithm in order to do off-policy optimization to learn a policy. One example, which we also use in our experiment, is the simple gradient ascent algorithm, given below in Algorithm 1. Here, $\mathrm{Proj}_\Theta$ is the projection onto $\Theta$.

---
**Algorithm 1** Off-policy projected gradient ascent
---
**Input:** An initial point $\theta_1 \in \Theta$ and step size schedule $\alpha_t$
**for** $t = 1, 2, \cdots$ **do**
  $\tilde{\theta}_{t+1} = \theta_t + \alpha_t \hat{Z}(\theta_t)$
  $\theta_{t+1} = \mathrm{Proj}_\Theta(\tilde{\theta}_{t+1})$
**end for**

---

## D  Proof

We often drop a time index $j$ for simplicity, like $p(s_j|s_{j-1}, a_{j-1}), p(r_j|s_j, a_j)$ instead of $p_j(s_j|s_{j-1}, a_{j-1}), p_j(r_j|s_j, a_j)$ for the ease of the notation.

### D.1  Proof of Theorem 1

We prove Theorem 1. Before that, we prove Theorem 6.

**Theorem 6.** *Let us define*

$$\phi_1(s, a, r; q, \pi^{\mathrm{b}}) = \frac{K_h(a - \tau(s))\{r - q(s,a)\}}{\pi^{\mathrm{b}}(a|s)} + v(s), v(s) = \int K_h(a - \tau(s))q(s,a)\mathrm{d}(a).$$

*In the case $\mathcal{K}$, assuming (b), we have*

$$\mathbb{E}[A_n] = o(1/\sqrt{nh}), \mathrm{var}[A_n] = o(1/(nh)) \tag{8}$$
$$A_n = \mathbb{P}_{\mathcal{U}_1}[\phi_1(S, A, R; \hat{q}^{[1]}, \hat{\pi}^{b[1]})] + \mathbb{P}_{\mathcal{U}_2}[\phi_1(S, A, R; \hat{q}^{[2]}, \hat{\pi}^{b[2]})] - \mathbb{P}_n[\phi_1(S, A, R; q, \pi^b)].$$

*In the case $\mathcal{D}$, the above (8) also holds assuming (a).*

#### D.1.1  Proof of Theorem 6, Case $\mathcal{K}$

What we will prove is

$$\mathbb{E}[\phi_1(S, A, R; \hat{q}^{[1]}, \hat{\pi}^{b[1]}) - \phi_1(S, A, R; q, \pi^b)|\mathcal{U}_2] = o((nh)^{-1/2}), \tag{9}$$
$$\mathbb{E}[\{\phi_1(S, A, R; \hat{q}^{[1]}, \hat{\pi}^{b[1]}) - \phi_1(S, A, R; q, \pi^b)\}^2|\mathcal{U}_2] = o(h^{-1}). \tag{10}$$

Then, the same argument holds for $\phi_1(S, A, R; \hat{q}^{[2]}, \hat{\pi}^{b[2]}) - \phi_1(S, A, R; q, \pi^b)$. The desired statement is concluded since

$$\mathbb{E}[\mathbb{P}_{\mathcal{U}_1}[\phi_1(S, A, R; \hat{q}^{[1]}, \hat{\pi}^{b[1]})] - \mathbb{P}_{\mathcal{U}_1}[\phi_1(S, A, R; q, \pi^b)]]$$
$$= \mathbb{E}[\mathbb{E}[\mathbb{P}_{\mathcal{U}_1}[\phi_1(S, A, R; \hat{q}^{[1]}, \hat{\pi}^{b[1]})] - \mathbb{P}_{\mathcal{U}_1}[\phi_1(S, A, R; q, \pi^b)] \mid \mathcal{U}_2]],$$
$$= \mathbb{E}[\mathbb{E}[\phi_1(S, A, R; \hat{q}, \hat{\pi}^b) - \phi_1(S, A, R; q, \pi^{b[1]}) \mid \mathcal{U}_2]].$$

In addition,

$$\mathbb{E}[\left\{\mathbb{P}_{\mathcal{U}_1}[\phi_1(S, A, R; \hat{q}^{[1]}, \hat{\pi}^{b[1]})] - \mathbb{P}_{\mathcal{U}_1}[\phi_1(S, A, R; q, \pi^b)]\right\}^2]$$
$$= \mathbb{E}[\mathbb{E}[\left\{\mathbb{P}_{\mathcal{U}_1}[\phi_1(S, A, R; \hat{q}^{[1]}, \hat{\pi}^{b[1]})] - \mathbb{P}_{\mathcal{U}_1}[\phi_1(S, A, R; q, \pi^b)]\right\}^2 |\mathcal{U}_2]],$$
$$= n^{-1}K\mathbb{E}[\mathbb{E}[\{\phi_1(S, A, R; \hat{q}^{[1]}, \hat{\pi}^{b[1]}) - \phi_1(S, A, R; q, \pi^b)\}^2 \mid \mathcal{U}_2]].$$

**Proof of Eq. (9) and Eq. (10)**  In this subsection, we remove $\{[1]\}$ for the ease of the notation. To prove (9), we show

$$\mathbb{E}\left[K_h(A - \tau(S))\left\{\frac{1}{\hat{\pi}^b(A|S)} - \frac{1}{\pi^b(A|S)}\right\}\{q(S,A) - \hat{q}(S,A)\}\right] = o(nh)^{-1/2}.$$

This is proved by

$$\mathbb{E}\left[\mathbb{E}\left[K_h(A - \tau(S))\left\{\frac{1}{\hat{\pi}^b(A|S)} - \frac{1}{\pi^b(A|S)}\right\}\{q(S,A) - \hat{q}(S,A)\} \mid \mathcal{U}_2\right]\right]$$

$$= \mathbb{E}\left[\int \frac{1}{h}k((a - \tau(s))h^{-1})\left\{\frac{1}{\hat{\pi}^b(a|s)} - \frac{1}{\pi^b(a|s)}\right\}\{q(s,a) - \hat{q}(s,a)\}\pi^b(a \mid s)p(s)\mathrm{d}(s,a)\right]$$

$$= \mathbb{E}\left[\int k(u)\left\{\frac{1}{\hat{\pi}^b(\tau(s) + uh|s)} - \frac{1}{\pi^b(\tau(s) + uh|s)}\right\}\{q(s,\tau(s) + uh) - \hat{q}(s,\tau(s) + uh)\}\pi^b(\tau(s) + uh \mid s)p(s)\mathrm{d}(u,s)\right]$$

$$= \mathbb{E}\Big[\int k(u)\left\{\frac{1}{\hat{\pi}^b(\tau(s)|s)} - \frac{1}{\pi^b(\tau(s)|s)} + uh\left\{-\frac{\hat{\pi}^{b(1)}(\tau(s)|s)}{\hat{\pi}^{2b}(\tau(s)|s)} + \frac{\pi^{b(1)}(\tau(s)|s)}{\pi^{2b}(\tau(s)|s)}\right\} + \mathcal{O}((uh^2))\right\} \times$$

$$\{q(s,\tau(s)) - \hat{q}(s,\tau(s)) + uh\{q^{(1)}(s,\tau(s)) - \hat{q}^{(1)}(s,\tau(s))\} + \mathcal{O}((uh^2))\}\times$$

$$\{\pi^b(\tau(s) \mid s) + uh\pi^{b(1)}(\tau(s) \mid s) + \mathcal{O}((uh^2))\}p(s)\mathrm{d}(u,s)\Big]$$

$$= M_2(k)\mathbb{E}\left[\int \left\{\frac{1}{\hat{\pi}^b(\tau(s) \mid s)} - \frac{1}{\pi^b(\tau(s) \mid s)}\right\}\{q(s,\tau(s)) - \hat{q}(s,\tau(s))\}\pi^b(\tau(s) \mid s)p(s)\mathrm{d}(s)\right] + \mathcal{O}(h^2) \times o(1)$$

$$= o((nh)^{-1/2}).$$

In the last line, we use the assumptions that $q(a,x), \pi^b(a|x), \hat{q}(a,x), \hat{\pi}^b(a|x)$ are $C^2$-functions wrt actions, and

$$\mathbb{E}\left[\left\|\frac{1}{\hat{\pi}^b(A \mid S)} - \frac{1}{\pi^b(A \mid S)}\right\|_\infty \|q(S,A) - \hat{q}(S,A)\|_\infty\right] = o(n^{-1/2}h^{-1/2}),$$

$$\mathbb{E}\left[\left\|\frac{1}{\hat{\pi}^b(A) \mid S)} - \frac{1}{\pi^b(A \mid S)}\right\|_{1,\infty}\right] = o(1), \mathbb{E}\left[\|\hat{q}(S,A) - \hat{q}(S,A)\|_{1,\infty}\right] = o(1),$$

$$\mathcal{O}(h^2) \times o(1) = o((nh)^{-1/2}), nh^5 = \mathcal{O}(1).$$

In addition, Eq. (10) is proved since

$$\mathbb{E}[\mathbb{E}[\{\phi_1(S,A,R;\hat{q}^{[1]}, \hat{\pi}^{b[1]}) - \phi_1(S,A,R;q,\pi^b)\}^2 \mid \mathcal{U}_2]]$$

$$\lesssim \mathbb{E}\left[\mathbb{E}\left[K_h(A - \tau(S))^2\left\{\frac{1}{\hat{\pi}^b(A \mid S)} - \frac{1}{\pi^b(A \mid S)}\right\}^2 \{q(S,A) - \hat{q}(S,A)\}^2 \mid \mathcal{U}_2\right]\right]$$

$$+ \mathbb{E}\left[\mathbb{E}\left[K_h(A - \tau(S))^2\left\{\frac{1}{\hat{\pi}^b(A \mid S)} - \frac{1}{\pi^b(A \mid S)}\right\}^2 \{R - q(S,A)\}^2 \mid \mathcal{U}_2\right]\right]$$

$$+ \mathbb{E}\left[\mathbb{E}\left[\frac{K_h(A - \tau(S))^2}{\pi^b(A \mid S)^2}\{q(S,A) - \hat{q}(S,A)\}^2 \mid \mathcal{U}_2\right] + \mathbb{E}\left[\{\hat{v}(S) - v(S)\}^2 \mid \mathcal{U}_2\right]\right]$$

$$\lesssim h^{-1} \max\left\{\mathbb{E}[\|\hat{q}(S,\tau(S)) - q(S,\tau(S))\|_2^2], \mathbb{E}\left[\left\|\frac{1}{\hat{\pi}^b(\tau(S)|S)} - \frac{1}{\pi^b(\tau(S)|S)}\right\|_2^2\right]\right\} + \mathcal{O}(1) = o(h^{-1}).$$

#### D.1.2  Proof of Theorem 6, Case $\mathcal{D}$

Essentially, the same proof is seen in Colangelo and Lee (2019). For completeness, we also write the proof here with our notation. Let us define

$$\phi_2(s,a,r;q,\pi^b) = \frac{K_h(a - \tau(s))\{r - q(s,\tau(s))\}}{\pi^b(a|s)} + q(S,\tau(S)).$$

As in the argument similar to the case $\mathcal{K}$, what we have to prove is

$$\mathbb{E}[\phi_2(S,A,R;\hat{q}^{[1]}, \hat{\pi}^{b[1]}) - \phi_2(S,A,R;q,\pi^b)|\mathcal{U}_2] = o((nh)^{-1/2}), \tag{11}$$

$$\mathbb{E}[\{\phi_2(S,A,R;\hat{q}^{[1]}, \hat{\pi}^{b[1]}) - \phi_2(S,A,R;q,\pi^b)\}^2|\mathcal{U}_2] = o(h^{-1}). \tag{12}$$

In this subsection, we remove $\{[1]\}$ for the ease of the notation.

Eq. (11) is proved since

$$\mathbb{E}\left[\mathbb{E}[\phi_2(S,A,R;\hat{q},\hat{\pi}^b) - \phi_2(S,A,R;q,\pi^b) \mid \mathcal{U}_2]\right]$$

$$= \mathbb{E}\left[\mathbb{E}\left[K_h(A-\tau(S))\left\{\frac{1}{\hat{\pi}^b(\tau(S)\mid S)} - \frac{1}{\pi^b(\tau(S)\mid S)}\right\}\{q(S,\tau(S)) - \hat{q}(S,\tau(S))\} \mid \mathcal{U}_2\right]\right] + \tag{13}$$

$$+ \mathbb{E}\left[\mathbb{E}\left[K_h(A-\tau(S))\left\{\frac{1}{\hat{\pi}^b(\tau(S)\mid S)} - \frac{1}{\pi^b(\tau(S)\mid S)}\right\}\{R - q(S,\tau(S))\} \mid \mathcal{U}_2\right]\right] \tag{14}$$

$$+ \mathbb{E}\left[\mathbb{E}\left[\frac{K_h(A-\tau(S))}{\pi^b(\tau(S)\mid S)}\{q(S,\tau(S)) - \hat{q}(S,\tau(S))\} + \hat{q}(S,\tau(S)) - q(S,\tau(S)) \mid \mathcal{U}_2\right]\right] \tag{15}$$

$$= o((nh)^{-1/2}) + o(1) \times \mathcal{O}(h^2) + o(1) \times \mathcal{O}(h^2) = o(nh)^{-1/2}.$$

Here, we use the facts that (13) is $o((nh)^{-1/2})$, (14) is $o(1) \times \mathcal{O}(h^2)$, (15) is $o(1) \times \mathcal{O}(h^2)$, which we will prove soon. In the last line, we use $nn^5 = \mathcal{O}(1)$. From now on, we prove (14) is $o(1) \times \mathcal{O}(h^2)$:

$$\mathbb{E}\left[\mathbb{E}\left[K_h(A-\tau(S))\left\{\frac{1}{\hat{\pi}^b(\tau(S)\mid S)} - \frac{1}{\pi^b(\tau(S)\mid S)}\right\}\{R - q(S,\tau(S))\} \mid \mathcal{U}_2\right]\right]$$

$$= \mathbb{E}\left[\mathbb{E}\left[\left\{\frac{1}{\hat{\pi}^b(\tau(S)\mid S)} - \frac{1}{\pi^b(\tau(S)\mid S)}\right\}\{\mathbb{E}[K_h(A-\tau(S))q(S,A) \mid S] - K_h(A-\tau(S))q(S,\tau(S))\} \mid \mathcal{U}_2\right]\right]$$

$$= \mathbb{E}\left[\mathbb{E}\left[\left\{\frac{1}{\hat{\pi}^b(\tau(S)\mid S)} - \frac{1}{\pi^b(\tau(S)\mid S)}\right\}\{\mathcal{O}(h^2)\} \mid \mathcal{U}_2\right]\right]$$

$$= o(1) \times \mathcal{O}(h^2).$$

More specifically,

$$\mathbb{E}[K_h(A-\tau(S))\{q(S,A) - q(S,\tau(S))\} \mid S] = \int \frac{1}{h}k\left\{\frac{a-\tau(s)}{h}\right\}\pi^b(a\mid s)\{q(s,a) - q(s,\tau(s))\}\mathrm{d}a$$

$$= \int k(u)\pi^b(\tau(s) + uh\mid s)\{q(s,\tau(s) + uh) - q(s,\tau(s))\}\mathrm{d}u$$

$$= \int k(u)\{\pi^b(\tau(s)\mid s) + \mathcal{O}(uh)\}\{uhq^{(1)}(s,\tau(s)) + \mathcal{O}(h^2)\}\mathrm{d}u = \mathcal{O}(h^2).$$

noting $\int uk(u)\mathrm{d}u = 0$. Next, we prove (15) is $o(1) \times \mathcal{O}(h^2)$:

$$\mathbb{E}\left[\mathbb{E}\left[\frac{K_h(A-\tau(S))}{\pi^b(\tau(S)\mid S)}\{q(S,\tau(S)) - \hat{q}(S,\tau(S))\} + \hat{q}(S,\tau(S)) - q(S,\tau(S)) \mid \mathcal{U}_2\right]\right]$$

$$= \mathbb{E}\left[\mathbb{E}\left[\left\{\frac{K_h(A-\tau(S))}{\pi^b(\tau(S)\mid S)} - 1\right\}\{q(S,\tau(S)) - \hat{q}(S,\tau(S))\} \mid \mathcal{U}_2\right]\right]$$

$$= \mathbb{E}\left[\mathbb{E}\left[\{\mathcal{O}(h^2)\}\{q(S,\tau(S)) - \hat{q}(S,\tau(S))\} \mid \mathcal{U}_2\right]\right] = o(1) \times \mathcal{O}(h^2).$$

Eq. (12) is similarly proved as in the case $\mathcal{K}$.

### D.1.3  Proof of Theorem 1

We prove the statement for the case $\mathcal{K}$. The statement for the case $\mathcal{D}$ is similarly proved as in (Colangelo and Lee, 2019).

**Bias term**  The bias term is calculated as follows:

$$\mathbb{E}[\mathbb{P}_n[\phi_1(S,A,R;q,\pi^b)]] - J$$

$$= \mathbb{E}\left[\int K_h(a-\tau(S))q(S,a)\mathrm{d}a\right] - J$$

$$= \mathbb{E}\left[\int k(u)\{q(S,\tau(S) + uh) - q(S,\tau(S))\}\mathrm{d}u\right]$$

$$= 0.5h^2\mathbb{E}\left[\int k(u)u^2q^{(2)}(S,\tau(S))\mathrm{d}u\right] + o(h^2) = 0.5h^2M_2(k)\mathbb{E}\left[q^{(2)}(S,\tau(S))\right] + o(h^2).$$

Here, we use a smoothness assumption. More formally, from the third line to the fourth line, based on the function $a \to q(s,a)$ is a $C^2$-function on the compact space, we use

$$q(s, \tau(s) + uh) - q(s, \tau(s)) = uhq^{(1)}(s, \tau(s)) + 0.5(uh)^2 q^{(2)}(s, \tau(s)) + o((uh)^2).$$

Refer to Li and Racine (2007, Exericise 1.5). Then, the all of the bias is

$$\mathbb{E}[\mathbb{P}_n[\phi_1(S, A, R; q, \pi^b)]] - J = \mathbb{E}[\mathbb{P}_n[\phi_1(S, A, R; q, \pi^b)]] - J + o(n^{-1/2}h^{-1/2})$$
$$= 0.5h^2 M_2(k)\mathbb{E}\left[q^{(2)}(S, \tau(S))\right] + o(n^{-1/2}h^{-1/2}).$$

Finally, the statement is concluded since

$$\mathbb{E}[\hat{J}^{\mathcal{K}}] - J = \mathbb{E}[\mathbb{P}_n[\phi_1(S, A, R; q, \pi^b)]] - J + o((nh)^{-1/2})$$
$$= 0.5h^2 M_2(k)\mathbb{E}\left[q^{(2)}(S, \tau(S))\right] + o(n^{-1/2}h^{-1/2}).$$

**Variance term**   The variance term is calculated as follows. First, we have

$$\mathrm{var}[\mathbb{P}_n[\phi_1(S, A, R; q, \pi^b)]]$$
$$= \frac{1}{n}\left(\mathbb{E}[\phi_1(S, A, R; q, \pi^b)^2] - \{\mathbb{E}[\phi_1(S, A, R; q, \pi^b)]\}^2\right)$$
$$= \frac{1}{nh^2}[\int\left\{\frac{k((a - \tau(s))h^{-1})}{\pi^b(a \mid s)}\right\}^2\{r - q(s,a)\}^2 p(r \mid a, s)\pi^b(a \mid s)p(s)\mathrm{d}(a, s, r)$$
$$+ \int\left\{\int k((a - \tau(s))h^{-1})q(s,a)\mathrm{d}a\right\}^2 p(s)\mathrm{d}(s) + O(h^2)]$$
$$= \frac{1}{nh^2}\left[\int h\left\{\frac{k^2(u)}{\pi^b(\tau(s) + uh \mid s)}\right\}\{r - q(s, \tau(s) + uh)\}^2 p(r \mid s, \tau(s) + uh)p(s)\mathrm{d}(u, s, r) + O(h^2)\right]$$
$$= \frac{1}{nh^2}\left[\int h\left\{\frac{k^2(u)}{\pi^b(\tau(s) \mid s)}\right\}\{r - q(s, \tau(s))\}^2 p(r \mid s, \tau(s))p(s)\mathrm{d}(u, s, r) + O(h^2)\right]$$
$$= \frac{1}{nh}\{\Omega_2(k)V + o(h)\}, \quad V = \int\left\{\frac{\{r - q(s, \tau(s))\}^2}{\pi^b(\tau(s) \mid s)}\right\}p(r \mid s, \tau(s))p(s)\mathrm{d}(s, r).$$

Here, we use smoothness assumptions, and $\left\{\int k((a - \tau(s))h^{-1})q(s,a)\mathrm{d}a\right\}^2 = O(h^2)$, which is proved by a standard algebra. Then,

$$\mathrm{var}[\mathbb{P}_n[\phi_1(S, A, R; \hat{q}, \hat{\pi}^b)]]$$
$$= \mathrm{var}[\mathbb{P}_n[\phi_1(S, A, R; q, \pi^b)]] + \mathrm{var}[\mathbb{P}_n[\phi_1(S, A, R; q, \pi^b) - \phi_1(S, A, R; \hat{q}, \hat{\pi}^b)]]$$
$$+ 2\{\mathrm{var}[\mathbb{P}_n[\phi_1(S, A, R; q, \pi^b)]]\mathrm{var}[\mathbb{P}_n[\phi_1(S, A, R; q, \pi^b) - \phi_1(S, A, R; \hat{q}, \hat{\pi}^b)]]\}^{1/2}$$
$$= \frac{\Omega_2(k)}{nh}\{V + o(h)\} + 2\{\frac{1}{nh}\{V + o(h)\}\}^{1/2}o(n^{-1/2}h^{-1/2}) + o(n^{-1}h^{-1})$$
$$= \frac{\Omega_2(k)}{nh}\{V + o(1)\}.$$

**Remark 9.** Colangelo and Lee (2019, Theorem 1) showed that the constant in the bias term of case $\mathcal{D}$ is

$$\mathbb{E}[0.5q^{(2)}(s, \tau(s)) + q^{(1)}(s, \tau(s))\pi^{b(1)}(a|s)/\pi^b(a|s)].$$

## D.2   Proof of Corollary 1

Obvious from Theorem 1.

## D.3   Proof of Theorem 2

**Replacing parts**  Let us define

$$\psi_1^{\mathcal{K}}(s,a,r;q,\pi^{\mathrm{b}}) = \left\{ \frac{\nabla K_h(a-\tau(s))\{r-q(s,a)\}}{\pi^{\mathrm{b}}(a|s)} + \int \nabla K_h(a-\tau(s))q(s,a)\mathrm{d}(a) \right\}.$$

Here, we prove that nuisance estimators can be replaced with true functions in the sense that

$$\mathbb{E}[A_n] = o(n^{-1/2}h^{-3/2}),\ \mathbb{E}[A_n^2] = o(n^{-1}h^{-3}),$$

where

$$A_n = \mathbb{P}_{\mathcal{U}_1}[\psi_1^{\mathcal{K}}(S,A,R;\hat{q}^{[1]},\hat{\pi}^{\mathrm{b}[1]})|\mathcal{U}_2] + \mathbb{P}_{\mathcal{U}_2}[\psi_1^{\mathcal{K}}(S,A,R;\hat{q}^{[2]},\hat{\pi}^{\mathrm{b}[2]})|\mathcal{U}_1] - \mathbb{P}_n[\psi_1^{\mathcal{K}}(S,A,R;q,\pi^{\mathrm{b}})].$$

Then, what we have to prove is

$$\mathbb{E}[\mathbb{E}[\psi_1^{\mathcal{K}}(S,A,R;\hat{q}^{[1]},\hat{\pi}^{\mathrm{b}[1]}) - \psi_1^{\mathcal{K}}(S,A,R;q,\pi^{b}) \mid \mathcal{U}_2]] = o(n^{-1/2}h^{-3/2}), \qquad (16)$$

$$\mathbb{E}[\mathbb{E}[\{\psi_1^{\mathcal{K}}(S,A,R;\hat{q}^{[1]},\hat{\pi}^{\mathrm{b}[1]}) - \psi_1^{\mathcal{K}}(S,A,R;q,\pi^{b})\}^2 \mid \mathcal{U}_2]] = o(h^{-3}). \qquad (17)$$

In this subsection, we remove $\{[1]\}$ for the ease of the notation. We write $1/\pi^{\mathrm{b}}(A|S)$ as $\eta(A|S)$. Eq. (16) is proved as follows:

$$\mathbb{E}[\mathbb{E}[\psi_1^{\mathcal{K}}(S,A,R;\hat{q},\hat{\pi}^b) - \psi_1^{\mathcal{K}}(S,A,R;q,\pi^b) \mid \mathcal{U}_2]] \qquad (18)$$

$$= \mathbb{E}[\mathbb{E}\left[\nabla K_h(A-\tau(S))\{\hat{\eta}(A|S) - \eta(A|S)\}\{q(S,A) - \hat{q}(S,A)\} \mid \mathcal{U}_2]]\right.$$

$$= \mathbb{E}\left[\mathbb{E}\left[-h^{-2}k^{(1)}\left(\frac{A-\tau(S)}{h}\right)\nabla\tau(S)\{\hat{\eta}(A|S) - \eta(A|S)\}\{q(S,A) - \hat{q}(S,A)\} \mid \mathcal{U}_2\right]\right]$$

$$= \mathbb{E}\left[\mathbb{E}\left[-h^{-2}k^{(1)}\left(\frac{A-\tau(S)}{h}\right)\nabla\tau(S)\{\hat{\eta}(A|S) - \eta(A|S)\}\{q(S,A) - \hat{q}(S,A)\} \mid \mathcal{U}_2\right]\right]$$

$$= \mathbb{E}[\mathbb{E}[\int -h^{-1}k^{(1)}(u)\nabla\tau(S)\{\hat{\eta}(\tau(S) + uh|S) - \eta(\tau(S) + uh|S)\} \times$$

$$\{q(S,\tau(S) + uh) - \hat{q}(S,\tau(S) + uh)\}\pi^{\mathrm{b}}(\tau(s) + uh|S)\mathrm{d}u \mid \mathcal{U}_2]] \qquad (19)$$

$$= \mathbb{E}[\mathbb{E}[\int k(u)\nabla\tau(S)\left\{\hat{\eta}^{(1)}(\tau(S) + uh|S) - \eta^{(1)}(\tau(S) + uh|S)\right\} \times$$

$$\{q(S,\tau(S) + uh) - \hat{q}(S,\tau(S) + uh)\}\pi^b(\tau(S) + uh|S)\mathrm{d}u \mid \mathcal{U}_2]] \qquad (20)$$

$$+ \mathbb{E}[\mathbb{E}[\int k(u)\nabla\tau(S)\{\hat{\eta}(\tau(S) + uh|S) - \eta(\tau(S) + uh|S)\} \times$$

$$\{q^{(1)}(S,\tau(S) + uh) - \hat{q}^{(1)}(S,\tau(S) + uh)\}\pi^b(\tau(S) + uh|S)\mathrm{d}u \mid \mathcal{U}_2]]$$

$$+ \mathbb{E}[\mathbb{E}[\int k(u)\nabla\tau(S)\{\hat{\eta}(\tau(S) + uh|S) - \eta(\tau(S) + uh|S)\} \times$$

$$\{q(S,\tau(S) + uh) - \hat{q}(S,\tau(S) + uh)\}\pi^{b(1)}(\tau(S) + uh|S)\mathrm{d}u \mid \mathcal{U}_2]]$$

Then, this is equal to

$$M_2(k)\mathbb{E}[\mathbb{E}[\nabla\tau(S)\left\{\hat{\eta}^{(1)}(\tau(S)|S) - \eta^{(1)}(\tau(S)|S)\right\}\{q(S,\tau(S)) - \hat{q}(S,\tau(S))\}\pi^{\mathrm{b}}(\tau(S)|S) \mid \mathcal{U}_2]] \quad (21)$$

$$+ M_2(k)\mathbb{E}[\mathbb{E}[\nabla\tau(S)\{\hat{\eta}(\tau(S)|S) - \eta(\tau(S)|S)\}\{q^{(1)}(S,\tau(S)) - \hat{q}^{(1)}(S,\tau(S))\}\pi^{\mathrm{b}}(\tau(S)|S) \mid \mathcal{U}_2]]$$

$$+ M_2(k)\mathbb{E}[\mathbb{E}[\nabla\tau(S)\{\hat{\eta}(\tau(S)|S) - \eta(\tau(S)|S)\}\{q(S,\tau(S)) - \hat{q}(S,\tau(S))\}\pi^{\mathrm{b}}(\tau(S)|S) \mid \mathcal{U}_2]]$$

$$+ o(h^2)$$

$$\lesssim \mathbb{E}[\|\|\nabla\tau(S)\|_{\mathrm{op}}\|_\infty\|\hat{q}(S,A) - q(S,A)\|_{1,\infty}\|\hat{\pi}^b(S,A) - \pi^b(S,A)\|_{1,\infty}] + o(n^{-1/2}h^{-3/2}) \qquad (22)$$

$$= o(n^{-1/2}h^{-3/2}).$$

Here, from (19) to (20), we have used a partial integration. From (20) to (21), we have used $a \to \eta(s,a)$ and $a \to q(s,a)$ are $C^3$-functions, and

$$\mathbb{E}\left[\left\{\hat{\eta}^{(1)}(\tau(S) + uh|S) - \eta^{(1)}(\tau(S) + uh|S)\right\}\{q(S,\tau(S) + uh) - \hat{q}(S,\tau(S) + uh)\} \mid \mathcal{U}_2\right]$$

$$= \mathbb{E}[\{\hat{\eta}^{(1)}(\tau(S)|S) - \eta^{(1)}(\tau(S)|S) + uh\{\hat{\eta}^{(2)}(\tau(S)|S) - \eta^{(2)}(\tau(S)|S)\} + \mathcal{O}((uh)^2)\} \times$$

$$\{q(S,\tau(S)) - \hat{q}(S,\tau(S)) + uh\{q^{(1)}(S,\tau(S)) - \hat{q}^{(1)}(S,\tau(S))\} + \mathcal{O}((uh)^2)\} \mid \mathcal{U}_2]$$

$$\lesssim \|\hat{\eta}(A|S) - \eta(A|S)\|_{1,\infty} \times \|\hat{q}(A|S) - q(A|S)\|_{1,\infty}$$

$$+ \{\|\hat{\eta}(A|S) - \eta(A|S)\|_{2,\infty}\|\hat{q}(A|S) - q(A|S)\|_{2,\infty} + \|\hat{\eta}(A|S) - \eta(A|S)\|_{1,\infty} + \|\hat{\eta}(A|S) - \eta(A|S)\|_{1,\infty}\} \times \mathcal{O}(h^2).$$

From (21) to (22), we use an assumption $nh^7 = \mathcal{O}(1)$.

Eq. (17) is proved as follows:

$$
\mathbb{E}\left[\mathbb{E}[\{\psi_1^{\mathcal{K}}(S, A, R; \hat{q}, \hat{\eta}) - \psi_1^{\mathcal{K}}(S, A, R; q, \eta)\}^2 \mid \mathcal{U}_2]\right]
$$

$$
\lesssim \mathbb{E}\left[\mathbb{E}\left[h^{-2}K_h^{(1)}(A - \tau(S))^2 \left\{\frac{1}{\hat{\pi}^b(A \mid S)} - \frac{1}{\pi^b(A \mid S)}\right\}^2 \{q(S, A) - \hat{q}(S, A)\}^2 \{\nabla\tau(S)\}^2 \mid \mathcal{U}_2\right]\right]
$$

$$
+ \mathbb{E}\left[\mathbb{E}\left[h^{-2}K_h^{(1)}(A - \tau(S))^2 \left\{\frac{1}{\hat{\pi}^b(A \mid S)} - \frac{1}{\pi^b(A \mid S)}\right\}^2 \{R - q(S, A)\}^2 \{\nabla\tau(S)\}^2 \mid \mathcal{U}_2\right]\right]
$$

$$
+ \mathbb{E}\left[\mathbb{E}\left[h^{-2}\frac{K_h^{(1)}(A - \tau(S))^2}{\pi^b(A \mid S)^2} \{q(S, A) - \hat{q}(S, A)\}^2 \{\nabla\tau(S)\}^2 \mid \mathcal{U}_2\right]\right]
$$

$$
+ \mathbb{E}\left[\mathbb{E}\left[h^{-2} \left\{\int K_h^{(1)}(a - \tau(S))\hat{q}(S, a)\mathrm{d}a - \int K_h^{(1)}(a - \tau(S))q(S, a)\mathrm{d}a\right\}^2 \{\nabla\tau(S)\}^2 \mid \mathcal{U}_2\right]\right]
$$

$$
\lesssim h^{-3} \times \mathbb{E}[\max\{\|\hat{\eta}(S, \tau(S)) - \eta(S, \tau(S))\|_2^2, \|\hat{\eta}^{(1)}(S, \tau(S)) - \eta^{(1)}(S, \tau(S))\|_2^2,
$$

$$
\|\hat{q}(S, \tau(S)) - q(S, \tau(S))\|_2^2, \|\hat{q}^{(1)}(S, \tau(S)) - q^{(1)}(S, \tau(S))\|_2^2\}] + o(h^{-2})
$$

$$
= o(h^{-3}).
$$

**Calculation of the bias and variance term**    The bias term is calculated as

$$
\mathbb{E}[\mathbb{P}_n[\psi_1^{\mathcal{K}}(S, A, R; q, \pi^b)]] - Z(\theta)
$$

$$
= \mathbb{E}\left[-\frac{\nabla\tau_\theta(S)}{h^2}\int k^{(1)}\left(\frac{a - \tau_\theta(S)}{h}\right)q(S, a)\mathrm{d}a\right] - Z(\theta)
$$

$$
= \mathbb{E}\left[-\frac{\nabla\tau_\theta(S)}{h}\int k^{(1)}(u)\, q(S, \tau_\theta(S) + uh)\mathrm{d}u\right] - Z(\theta)
$$

$$
= \mathbb{E}\left[\nabla\tau_\theta(S)\int k(u)\, q^{(1)}(S, \tau_\theta(S) + uh)\mathrm{d}u\right] - Z(\theta)
$$

$$
= \mathbb{E}\left[\nabla\tau_\theta(S)\int k(u)\, \{q^{(1)}(S, \tau_\theta(S) + uh) - q^{(1)}(S, \tau_\theta(S))\}\mathrm{d}u\right]
$$

$$
= 0.5h^2\mathbb{E}\left[\nabla\tau_\theta(S)q^{(3)}(S, \tau_\theta(S))\right]\int u^2k(u)\mathrm{d}u + o(h^2).
$$

In the last line, we have used that the function $a \to q(s, a)$ is a $C^3$-function. In addition, we also use a fact $nh^7 = \mathcal{O}(1)$ to say

$$
\hat{Z} - Z(\theta) = \mathbb{E}[\mathbb{P}_n[\psi_1^{\mathcal{K}}(S, A, R; q, \pi^b)]] - Z(\theta) + o(h^2)
$$

$$
= 0.5h^2\mathbb{E}\left[\nabla\tau_\theta(S)q^{(3)}(S, \tau_\theta(S))\right]\int u^2k(u)\mathrm{d}u + o(n^{-1/2}h^{-3/2}).
$$

**Remark 10.** Heuristically, this $0.5\mathbb{E}\left[\nabla\tau_\theta(S)q^{(3)}(S, \tau_\theta(S))\right]$ is calculated by differentiating the bias term of the OPE estimator: $0.5\mathbb{E}\left[q^{(2)}(S, \tau_\theta(S))\right]$.

The variance term is calculated as

$$\mathrm{var}[\mathbb{P}_n[\psi_1^{\mathcal{K}}(S, A, R; q, \pi^b)]] = \frac{1}{n}\mathbb{E}\left[\psi_1^{\mathcal{K}}(S, A, R; q, \pi^b)^2\right] + \circ(\frac{1}{nh^3})$$

$$= \frac{1}{n}\mathbb{E}\left[\otimes\nabla\tau_\theta(S)\frac{\{R - q(S, A)\}^2}{\{\pi^b(A \mid S)\}^2}K_h^{(1)}(A - \tau_\theta(S))^2\right] + \circ(\frac{1}{nh^3})$$

$$= \frac{1}{n}\mathbb{E}[\otimes\nabla\tau_\theta(S)\int\frac{\{r - q(S, a)\}^2}{\pi^b(a \mid S)}K_h^{(1)}(a - \tau_\theta(S))^2 p(r \mid S, a)\mathrm{d}(a, r)]$$

$$= \frac{1}{nh^4}\mathbb{E}[\otimes\nabla\tau_\theta(S)\int\frac{\{r - q(S, a)\}^2}{\pi^b(a \mid S)}k^{(1)}\left(\frac{a - \tau_\theta(S)}{h}\right)^2 p(r \mid S, a)\mathrm{d}(a, r)] + \circ(\frac{1}{nh^3})$$

$$= \frac{1}{nh^3}\mathbb{E}[\otimes\nabla\tau_\theta(S)\int\frac{\{r - q(S, \tau_\theta(S) + hu)\}^2}{\pi^b(\tau_\theta(S) + hu \mid s)}k^{(1)}(u)^2 p(r \mid S, \tau_\theta(S) + hu)\mathrm{d}(u, r)] + \circ(\frac{1}{nh^3})$$

$$= \frac{1}{nh^3}\left\{\mathbb{E}[\otimes\nabla\tau_\theta(S)\int\frac{\{r - q(S, \tau_\theta(S))\}^2}{\pi^b(\tau_\theta(S) \mid S)}k^{(1)}(u)^2 p(r \mid S, \tau_\theta(S))\mathrm{d}(u, r)] + \circ(1)\right\}$$

$$= \frac{1}{nh^3}\int k^{(1)}(u)^2\,\mathrm{d}(u)\left\{\mathbb{E}\left[\otimes\nabla\tau_\theta(S)\frac{\mathrm{var}[R \mid S, \tau_\theta(S)]}{\pi^b(\tau_\theta(S) \mid S)}\right] + \circ(1)\right\}.$$

Thus,

$$\mathrm{var}[\mathbb{P}_n[\psi_1^{\mathcal{K}}(S, A, R; \hat{q}, \hat{\pi}^b)]] = \mathrm{var}[\mathbb{P}_n[\psi_1^{\mathcal{K}}(S, A, R; q, \pi^b)]] + \circ(n^{-1}h^{-3})$$

$$= \frac{1}{nh^3}\Omega_2^{(1)}(k)\left\{\mathbb{E}\left[\otimes\nabla\tau_\theta(S)\frac{\mathrm{var}[R \mid S, \tau_\theta(S)]}{\pi^b(\tau_\theta(S) \mid S)}\right] + \circ(1)\right\}.$$

### D.4 Proof of Theorem 3

**Replacing estimators with true functions** We define

$$\phi^{\mathcal{K}}(\pi^b, q^{\mathcal{K}}) = v_0^{\mathcal{K}} + \sum_{t=1}^{H}\lambda_t^{\mathcal{K}}\{r_t - q_t^{\mathcal{K}} + v_{t+1}^{\mathcal{K}}\}.$$

Here, we prove that nuisance estimators can be replace with true functions:

$$\mathbb{E}[A_n] = \circ((nh^H)^{-1/2}), \quad \mathrm{var}[A_n] = \circ((nh^H)^{-1}),$$

$$A_n = \mathbb{P}_{\mathcal{U}_1}[\phi^{\mathcal{K}}(\hat{\pi}^{b[1]}, \hat{q}^{\mathcal{K}[1]})|\mathcal{U}_2] + \mathbb{P}_{\mathcal{U}_2}[\phi^{\mathcal{K}}(\hat{\pi}^{b[2]}, \hat{q}^{\mathcal{K}[2]})|\mathcal{U}_1] - \mathbb{E}[\phi^{\mathcal{K}}(\pi^b, q^{\mathcal{K}})].$$

Then, what we have to prove is

$$\mathbb{E}[\mathbb{E}[\phi^{\mathcal{K}}(\hat{\pi}^b, \hat{q}^{\mathcal{K}}) - \phi^{\mathcal{K}}(\pi^b, q^{\mathcal{K}})|\mathcal{U}_2]] = \circ((nh^H)^{-1/2}),$$

$$\mathbb{E}[\mathbb{E}[\{\phi^{\mathcal{K}}(\hat{\pi}^b, \hat{q}^{\mathcal{K}}) - \phi^{\mathcal{K}}(\pi^b, q^{\mathcal{K}})\}^2|\mathcal{U}_2]] = \circ(h^{-H}).$$

The rest of the part is proved as Theorem 4. Therefore, we omit the proof here.

Next, we analyze the bias and variance.

**Bias part** First, we have

$$\mathbb{E}[\mathbb{P}_n[\phi^{\mathcal{K}}(\pi^b, q^{\mathcal{K}})]] = \mathbb{E}[\textstyle\sum_t\lambda_t^{\mathcal{K}}r_t] - J(\theta) = \sum_t\mathbb{E}[\{\lambda_t^{\mathcal{K}} - \lambda_t\}r_t].$$

Here, we use a doubly robust property of $\phi^{\mathcal{K}}$. Then, by defining $c = q_t(s_t, a_t)$, the above is equal to

$$\sum_t\left\{\int\mathbb{E}[R_t \mid S_t = s_t, A_t = a_t]\prod_{i=1}^{t}\frac{K_h(a_i - \tau(s_i))}{\pi^b(a_i \mid s_i)}\{\prod_{i=1}^{t}\pi^b(a_i \mid s_i)p(s_i \mid s_{i-1}, a_{i-1})\}\mathrm{d}(h_{a_t}) - \mathbb{E}[\lambda_t r_t]\right\}$$

$$= \sum_t\left\{\int c(s_t, \tau_t(s_t) + hu_t)\prod_{i=1}^{t}k(u_i)\prod_{i=1}^{t}p(s_i \mid s_{i-1}, \tau(s_{i-1}) + hu_i)\mathrm{d}(h_{a_t}^u) - \mathbb{E}[\lambda_t r_t]\right\}$$

$$= \sum_t\{\int\{c(s_t, \tau_t(s_t) + hu_t) - c(s_t, \tau_t(s_t))\}\prod_{i=1}^{t}k(u_i)\prod_{i=1}^{t}p(s_i \mid s_{i-1}, \tau(s_{i-1}) + hu_i)\mathrm{d}(h_{a_t}^u) -$$

$$+ \int c(s_t, \tau_t(s_t))\prod_{i=1}^{t}k(u_i)\prod_{i=1}^{t}\{p(s_i \mid s_{i-1}, \tau(s_{i-1}) + hu_i) - p(s_i \mid s_{i-1}, \tau(s_{i-1}))\}\mathrm{d}(h_{a_t}^u)\}$$

where $c(s_t, a_t) = \mathbb{E}[Y_t \mid s_t, a_t]$, $h_{a_t}^u = \{s_1, u_1, s_2, \cdots\}$, $h_{a_t}^{u\tau} = \{s_1, \tau(s_1) + hu_1, s_2, \cdots\}$, $h_{a_t}^\tau = \{s_1, \tau(s_1), s_2, \cdots\}$. Then, we have

$$\sum_t \{\int \{h^2 u_t^2 c_t^{(2)}(s_t, \tau_t(s_t))\} \left\{\prod_{i=1}^t k(u_i) p(s_i \mid s_{i-1}, \tau(s_{i-1}))\right\} \mathrm{d}(h_{a_t}^u) -$$

$$\int c(s_t, a_t) \left\{\prod_{i=1}^t k(u_i)\right\} \sum_{l=1}^{t-1} \{h^2 u_l^2 p^{(2)}(s_{l+1} \mid s_l, \tau(s_l)) \prod_{j \neq l} p(s_j \mid s_{j-1}, \tau(s_{j-1}))\} \mathrm{d}(h_{a_t}^u)\} + o(h^2).$$

Here, $h_{a_t}^{-u} = \{s_1, s_2, \cdots\}$. Finally, it is equal to

$$0.5 h^2 M_2(k) \sum_{t=1}^H \left\{\mathbb{E}_\tau\left[\frac{r_t p^{(2)}(r_t \mid s_t, \tau_t(s_t))}{p(r_t \mid s_t, \tau_t(s_t))}\right] + \sum_{l=1}^{t-1} \mathbb{E}_\tau\left[\frac{r_t p^{(2)}(s_{l+1} \mid s_l, \tau(s_l))}{p(s_{l+1} \mid s_l, \tau(s_l))}\right]\right\} + o(h^2).$$

Then, we have

$$\mathbb{E}[\phi^{\mathcal{K}}(\hat{\pi}^{\mathrm{b}}, \hat{q}^{\mathcal{K}})] = \mathbb{E}[\phi^{\mathcal{K}}(\pi^{\mathrm{b}}, q^{\mathcal{K}})] + o((nh^H)^{-1/2})$$

$$= 0.5 h^2 M_2(k) \sum_{t=1}^H \left\{\mathbb{E}_\tau\left[\frac{r_t p^{(2)}(r_t \mid s_t, \tau_t(s_t))}{p(r_t \mid s_t, \tau_t(s_t))}\right] + \sum_{l=1}^{t-1} \mathbb{E}_\tau\left[\frac{r_t p^{(2)}(s_{l+1} \mid s_l, \tau(s_l))}{p(s_{l+1} \mid s_l, \tau(s_l))}\right]\right\} + o(h^2) + o((nh^H)^{-1/2})$$

$$= 0.5 h^2 M_2(k) \sum_{t=1}^H \left\{\mathbb{E}_\tau\left[\frac{r_t p^{(2)}(r_t \mid s_t, \tau_t(s_t))}{p(r_t \mid s_t, \tau_t(s_t))}\right] + \sum_{l=1}^{t-1} \mathbb{E}_\tau\left[\frac{r_t p^{(2)}(s_{l+1} \mid s_l, \tau(s_l))}{p(s_{l+1} \mid s_l, \tau(s_l))}\right]\right\} + o((nh^H)^{-1/2}).$$

**Variance part** The variance is

$$\mathrm{var}[\mathbb{P}_n[\phi^{\mathcal{K}}(\pi^{\mathrm{b}}, q^{\mathcal{K}})]] = \frac{1}{n} \mathrm{var}[\phi^{\mathcal{K}}(\pi^{\mathrm{b}}, q^{\mathcal{K}})]]$$

$$= \frac{1}{n} \sum_{t=1}^H \mathbb{E}\left[\left\{\prod_{i=1}^t \frac{h^{-1} k(h^{-1}(A_i - S_i))}{\pi^{\mathrm{b}}(A_i \mid S_i)}\right\}^2 \mathrm{var}[R_t + v^{\mathcal{K}}(S_{t+1}) \mid A_t, S_t]\right]. \tag{23}$$

Here, $\forall f$, we have

$$\mathbb{E}\left[\left\{\prod_{i=1}^t \frac{h^{-1} k(h^{-1}(A_i - \tau(S_i)))}{\pi^{\mathrm{b}}(A_i \mid S_i)}\right\}^2 f(\mathcal{H}_{A_t})\right]$$

$$= \frac{1}{h^{2t}} \int \prod_{i=1}^t \frac{k^2(h^{-1}(a_i - \tau(s_i)))}{\pi^{\mathrm{b}}(a_i \mid s_i)} f(h_{a_t}) \{\prod_{i=1}^t p(s_i \mid s_{i-1}, a_{i-1})\} \mathrm{d}(h_{a_t})$$

$$= \frac{1}{h^t} \int \prod_{i=1}^t \frac{k^2(u_i)}{\pi^{\mathrm{b}}(hu_i + \tau_i(s_i) \mid s_i)} f(h_{a_t}^{u\tau}) \{\prod_{i=1}^t p(s_i \mid s_{i-1}, hu_{i-1} + \tau(s_{i-1}))\} \mathrm{d}(h_{a_t}^u)$$

$$= \frac{1}{h^t} \left\{\int \prod_{i=1}^t \frac{k^2(u_i)}{\pi^{\mathrm{b}}(\tau_i(s_i) \mid s_i)} f(h_{a_t}^{u\tau}) \{\prod_{i=1}^t p(s_i \mid s_{i-1}, \tau(s_{i-1}))\} \mathrm{d}(h_{a_t}^u) + o(1)\right\}$$

$$= \frac{1}{h^t} \left\{\prod_{i=1}^t \int k^2(u_i) \mathrm{d}(u_i)\right\} \left\{\int \prod_{i=1}^t \left\{\frac{1}{\pi^{\mathrm{b}}(\tau(s_i) \mid s_i)}\right\} f(h_{a_t}^\tau) \{\prod_{i=1}^t p(s_i \mid s_{i-1}, \tau(s_{i-1}))\} \mathrm{d}(h_{a_t}^\tau) + o(1)\right\}.$$

Therefore, (23) is

$$\frac{1}{nh^H} \left\{\left\{\prod_{i=1}^H \int k^2(u_i) \mathrm{d}(u_i)\right\} \mathbb{E}_\tau\left[\prod_{i=1}^H \left\{\frac{1}{\pi^{\mathrm{b}}(\tau_i(s_i) \mid s_i)}\right\} \mathrm{var}[r_H \mid s_H, \tau(s_H)]\right] + o(1)\right\},$$

Finally,

$$\mathrm{var}[\mathbb{P}_n[\phi^{\mathcal{K}}(\hat{\pi}^{\mathrm{b}}, \hat{q}^{\mathcal{K}})]] = \mathrm{var}[\mathbb{P}_n[\phi^{\mathcal{K}}(\pi^{\mathrm{b}}, q^{\mathcal{K}})]] + o(n^{-1} h^{-H})$$

$$= \frac{1}{nh^H} \left\{\Omega_2^H(k) \mathbb{E}_\tau\left[\prod_{i=1}^H \left\{\frac{1}{\pi^{\mathrm{b}}(\tau_i(s_i) \mid s_i)}\right\} \mathrm{var}[r_H \mid s_H, \tau(s_H)]\right] + o(1)\right\}.$$

## D.5 Proof of Theorem 4

**Replacing estimators with true functions** We define

$$\phi^{\mathcal{K}}(w^{\mathcal{K}}, \pi^{\mathrm{b}}, q^{\mathcal{K}}) = v_0^{\mathcal{K}} + \sum_{t=1}^{H} \frac{w_t^{\mathcal{K}} K_h(a_t - \tau_t(s_t))}{\pi^{\mathrm{b}}(a_t|s_t)} \{r_t - q_t^{\mathcal{K}} + v_{t+1}^{\mathcal{K}}\}.$$

Here, we prove that nuisance estimators can be replace with true functions:

$$\mathbb{E}[\mathbb{P}_{\mathcal{U}_1}[\phi^{\mathcal{K}}(\hat{w}^{\mathcal{K}[1]}, \hat{\pi}^{b[1]}, \hat{q}^{\mathcal{K}[1]})|\mathcal{U}_2] + \mathbb{P}_{\mathcal{U}_2}[\phi^{\mathcal{K}}(\hat{w}^{\mathcal{K}[2]}, \hat{\pi}^{b[2]}, \hat{q}^{\mathcal{K}[2]})|\mathcal{U}_1] - \mathbb{E}[\phi^{\mathcal{K}}(w^{\mathcal{K}}, \pi^{\mathrm{b}}, q^{\mathcal{K}})]] = o((nh)^{-1/2})$$

$$\mathbb{E}[\{\mathbb{P}_{\mathcal{U}_1}[\phi^{\mathcal{K}}(\hat{w}^{\mathcal{K}[1]}, \hat{\pi}^{b[1]}, \hat{q}^{\mathcal{K}[1]})|\mathcal{U}_2] + \mathbb{P}_{\mathcal{U}_2}[\phi^{\mathcal{K}}(\hat{w}^{\mathcal{K}[2]}, \hat{\pi}^{b[2]}, \hat{q}^{\mathcal{K}[2]})|\mathcal{U}_1] - \mathbb{E}[\phi^{\mathcal{K}}(w^{\mathcal{K}}, \pi^{\mathrm{b}}, q^{\mathcal{K}})]\}^2] = o((nh)^{-1/2}).$$

Then, what we have to prove is

$$\mathbb{E}[\mathbb{E}[\phi^{\mathcal{K}}(\hat{w}^{\mathcal{K}[1]}, \hat{\pi}^{b[1]}, \hat{q}^{\mathcal{K}[1]}) - \phi^{\mathcal{K}}(w^{\mathcal{K}}, \pi^{\mathrm{b}}, q^{\mathcal{K}})|\mathcal{U}_2]] = o((nh)^{-1/2}), \tag{24}$$

$$\mathbb{E}[\mathbb{E}[\{\phi^{\mathcal{K}}(\hat{w}^{\mathcal{K}[1]}, \hat{\pi}^{b[1]}, \hat{q}^{\mathcal{K}[1]}) - \phi^{\mathcal{K}}(w^{\mathcal{K}}, \pi^{\mathrm{b}}, q^{\mathcal{K}})\}^2|\mathcal{U}_2]] = o((h)^{-1}). \tag{25}$$

From now on, we omit $[1], \mathcal{U}_2$ for simplicity. By defining $\tilde{w}_t^{\mathcal{K}}(s_t, a_t) = w_t^{\mathcal{K}}(s_t)/\pi^{\mathrm{b}}(a_t|s_t)$, Eq. (24) is proved by

$$\mathbb{E}[\phi^{\mathcal{K}}(\hat{w}^{\mathcal{K}}, \hat{\pi}^b, \hat{q}^{\mathcal{K}}) - \phi^{\mathcal{K}}(w^{\mathcal{K}}, \pi^{\mathrm{b}}, q^{\mathcal{K}})|\mathcal{U}_2]$$

$$= \mathbb{E}[\sum_{t=1}^{H}(\hat{\tilde{w}}_t^{\mathcal{K}}(S_t, A_t) - \tilde{w}_t^{\mathcal{K}}(S_t, A_t))K_h((A_t - \tau(S_t))h^{-1})(-\hat{q}_t^{\mathcal{K}}(S_t, A_t) + q_t^{\mathcal{K}}(S_t, A_t) + \hat{v}_{t+1}^{\mathcal{K}}(S_{t+1}) - v_{t+1}^{\mathcal{K}}(S_{t+1}))] \tag{26}$$

$$= M_2(k)\mathbb{E}[\sum_{t=1}^{H} \left\{\hat{\tilde{w}}_t^{\mathcal{K}}(S_t, \tau(S_t)) - \tilde{w}_t^{\mathcal{K}}(S_t, \tau(S_t))\right\}\left\{-\hat{q}_t^{\mathcal{K}}(S_t, \tau(S_t)) + q_t^{\mathcal{K}}(S_t, \tau(S_t))\right\}\pi^{\mathrm{b}}(\tau(S_t)|S_t)] \tag{27}$$

$$+ M_2(k)\mathbb{E}[\sum_{t=1}^{H} \left\{\hat{\tilde{w}}_t^{\mathcal{K}}(S_t, \tau(S_t)) - \tilde{w}_t^{\mathcal{K}}(S_t, \tau(S_t))\right\}(\hat{v}_{t+1}^{\mathcal{K}}(S_{t+1}) - v_{t+1}^{\mathcal{K}}(S_{t+1}))\pi^{\mathrm{b}}(\tau(S_t)|S_t)] + o(h^2)$$

$$= M_2(k)\mathbb{E}[\sum_{t=1}^{H} \left\{\hat{\tilde{w}}_t^{\mathcal{K}}(S_t, \tau(S_t)) - \tilde{w}_t^{\mathcal{K}}(S_t, \tau(S_t))\right\}\left\{-\hat{q}_t^{\mathcal{K}}(S_t, \tau(S_t)) + q_t^{\mathcal{K}}(S_t, \tau(S_t))\right\}\pi^{\mathrm{b}}(\tau(S_t)|S_t)] \tag{28}$$

$$+ M_2(k)\mathbb{E}[\sum_{t=1}^{H} \left\{\hat{\tilde{w}}_t^{\mathcal{K}}(S_t, \tau(S_t)) - \tilde{w}_t^{\mathcal{K}}(S_t, \tau(S_t))\right\}\left\{\hat{q}_{t+1}^{\mathcal{K}}(S_{t+1}, \tau(S_{t+1})) - q_{t+1}^{\mathcal{K}}(S_{t+1}, \tau(S_{t+1}))\right\}\pi^{\mathrm{b}}(\tau(S_t)|S_t)] + o(h^2)$$

$$= M_2(k)\sum_{t=1}^{H} \mathbb{E}[\|\hat{\tilde{w}}_t^{\mathcal{K}} - \tilde{w}_t^{\mathcal{K}}\|_\infty \left\{\|\hat{q}_t^{\mathcal{K}} - q_t^{\mathcal{K}}\|_\infty + \|\hat{q}_{t+1}^{\mathcal{K}} - q_{t+1}^{\mathcal{K}}\|_\infty\right\}] + o(h^2) \tag{29}$$

$$= o(n^{-1/2}h^{-1/2}). \tag{30}$$

Here, from (26) to (27),

$$\mathbb{E}[\max\{\|\hat{\pi}^b - \pi^b\|_{1,\infty}, \|\hat{w}_j^{\mathcal{K}} - w_j^{\mathcal{K}}\|_\infty, \|\hat{q}_j^{\mathcal{K}} - q_j^{\mathcal{K}}\|_{1,\infty}\}] = o(1).$$

From (27) to (28), $v_t^{\mathcal{K}}(s) = q_t^{\mathcal{K}}(s, \tau(s)) + o(1)$. From (28) to (29), we use

$$\mathbb{E}[\max\{\|\hat{\pi}^b - \pi^b\|_\infty, \|\hat{w}_j^{\mathcal{K}} - w_j^{\mathcal{K}}\|_\infty\}\|\hat{q}_j^{\mathcal{K}} - q_j^{\mathcal{K}}\|_\infty] = o(n^{-1/2}h^{-1/2}).$$

From (29) to (30), we use $nh^5 = \mathcal{O}(1)$. Eq. (25) is proved by

$$\mathbb{E}[\mathbb{E}[\{\phi^{\mathcal{K}}(\hat{w}^{\mathcal{K}}, \hat{\pi}^b, \hat{q}^{\mathcal{K}}) - \phi^{\mathcal{K}}(w^{\mathcal{K}}, \pi^{\mathrm{b}}, q^{\mathcal{K}})\}^2|\mathcal{U}_2]]$$

$$= \mathbb{E}[h^{-1}\max\{\|\hat{q}_t^{\mathcal{K}}(S_t, \tau(S_t)) - q_t^{\mathcal{K}}(S_t, \tau(S_t))\|_2^2, \|\hat{\tilde{w}}^{\mathcal{K}}(S_t, \tau(S_t)) - \tilde{w}^{\mathcal{K}}(S_t, \tau(S_t))\|_2^2\}] = o(h^{-1}).$$

**Calculation of bias and variance term** Bias and variance terms are bounded as follows.

**Bias part**

Bias is the same as the proof of Theorem 3. It is reduced to

$$0.5h^2 M_2(k) \sum_{t=1}^{H} \left\{ \mathbb{E}_\tau[c^{(2)}(s_t, \tau_t(s_t))] + \sum_{j=1}^{t-1} \mathbb{E}_\tau \left[ \frac{r_j p^{(2)}(s_{j+1}|s_j, \tau_j(s_j))}{p(s_j|s_j, \tau_j(s_j))} \right] \right\} + o(n^{-1/2}h^{-1/2}).$$

where $c(s, a) = \mathbb{E}[r|s, a]$.

**Variance part**

The variance is

$$\text{var}[\mathbb{P}_n[\phi^{\mathcal{K}}(w^{\mathcal{K}}, \pi^{\mathrm{b}}, q^{\mathcal{K}})]] = \frac{1}{n}\text{var}[\phi^{\mathcal{K}}(w^{\mathcal{K}}, \pi^{\mathrm{b}}, q^{\mathcal{K}})]]$$

$$= \frac{1}{n}\sum_{t=0}^{H} \mathbb{E}\left[ \left\{ \mathbb{E}\left[ \prod_{i=1}^{t} \frac{h^{-1}k(h^{-1}(A_i - S_i))}{\pi^{\mathrm{b}}(A_i \mid S_i)} | S_t \right] \right\}^2 \left\{ \frac{h^{-1}k(h^{-1}(A_t - \tau(S_t)))}{\pi^{\mathrm{b}}(A_t \mid S_t)} \right\}^2 \text{var}[R_t + v^{\mathcal{K}}(S_{t+1}) \mid A_t, S_t] \right].$$

(31)

First, we have

$$\mathbb{E}\left[ \prod_{i=1}^{t-1} \frac{h^{-1}k(h^{-1}(A_i - \tau_i(S_i)))}{\pi^{\mathrm{b}}(A_i \mid S_i)} \mid S_t \right]$$

$$= \int \prod_{i=1}^{t-1} \frac{h^{-1}k(h^{-1}(a_i - \tau_i(s_i)))}{\pi^{\mathrm{b}}(a_i \mid s_i)} \left\{ \prod_{i=1}^{t-1} p(s_{i+1} \mid s_i, a_i)\pi^{\mathrm{b}}(a_i \mid s_i) \right\} \frac{p_1(s_1)}{p_{\pi^{\mathrm{b}}}(s_t)} \mathrm{d}(h_{a_{t-1}})$$

$$= \int \prod_{i=1}^{t-1} k(u_i) \prod_{i=1}^{t-1} p(s_{i+1} \mid s_i, \tau_i(s_i) + hu_i) \frac{p_1(s_1)}{p_{\pi^{\mathrm{b}}}(s_t)} \mathrm{d}(h_{a_{t-1}}^u)$$

$$= \left\{ \int \prod_{i=1}^{t-1} k(u_i) \prod_{i=1}^{t-1} p(s_{i+1} \mid s_i, \tau_i(s_i)) \frac{p_1(s_1)}{p_{\pi^{\mathrm{b}}}(s_t)} \mathrm{d}(h_{a_{t-1}}^u) \right\} + o(1) = \frac{p_{\tau_\theta}(s_t)}{p_{\pi^{\mathrm{b}}}(s_t)} + o(1).$$

Therefore, $\forall f$ we have

$$\mathbb{E}\left[ \left\{ \mathbb{E}\left[ \prod_{i=1}^{t} \frac{h^{-1}k(h^{-1}(A_i - \tau_i(S_i)))}{\pi^{\mathrm{b}}(A_i \mid S_i)} \mid S_t \right] \right\}^2 \left\{ \frac{h^{-1}k(h^{-1}(A_t - \tau(S_t)))}{\pi^{\mathrm{b}}(A_t \mid S_t)} \right\}^2 f(S_t, A_t) \right]$$

$$= \mathbb{E}\left[ \left\{ \frac{p_{\tau_\theta}(S_t)}{p_{\pi^{\mathrm{b}}}(S_t)} + o(1) \right\}^2 \left\{ \frac{h^{-1}k(h^{-1}(A_t - \tau(S_t)))}{\pi^{\mathrm{b}}(A_t \mid S_t)} \right\}^2 f(S_t, A_t) \right] + o(1)$$

$$= \frac{1}{h}\left\{ \int \frac{p_{\tau_\theta}^2(s_t)}{p_{\pi^{\mathrm{b}}}(s_t)} \left\{ \frac{k(u_t)}{\pi^{\mathrm{b}}(\tau_t(s_t) + u_t \mid s_t)} \right\}^2 f(s_t, a_t)\pi^{\mathrm{b}}(\tau_t(s_t) + u_t \mid s_t)\mathrm{d}(s_t, u_t) + o(1) \right\}$$

$$= \frac{1}{h}\left\{ \Omega_2(k) \int \frac{p_{\tau_\theta}^2(s_t)}{p_{\pi^{\mathrm{b}}}(s_t)\pi^{\mathrm{b}}(\tau_t(s_t) \mid s_t)} f(s_t, \tau_t(s_t))\mathrm{d}(s_t) + o(1) \right\}.$$

In addition, noting

$$v_t^{\mathcal{K}}(s_t) = \int h^{-1}k(h^{-1}(a_t - \tau_t(s_t)))q_t^{\mathcal{K}}(s_t, a_t)\mathrm{d}(a_t)$$

$$= \int k(u_t)q_t^{\mathcal{K}}(s_t, \tau_t(s_t))\mathrm{d}(u_t) + o(1) = q_t^{\mathcal{K}}(s_t, \tau_t(s_t)) + o(1),$$

by induction, we have $v_t^{\mathcal{K}}(s_t) = v_t^{\pi^{\mathrm{e}}}(s_t) + o(1) = q_t^{\pi^{\mathrm{e}}}(s_t, \tau_t(s_t)) + o(1)$. Then, noting

$$\text{var}[R_t + v_{t+1}^{\mathcal{K}}(S_{t+1}) \mid S_t = s_t, A_t = a_t] = \text{var}[R_t + v_{t+1}^{\pi^{\mathrm{e}}}(S_{t+1}) \mid S_t = s_t, A_t = a_t] + o(1).$$

Therefore, the variance term is

$$\text{var}[\mathbb{P}_n[\phi^{\mathcal{K}}(\hat{w}^{\mathcal{K}}, \hat{\pi}^{\mathrm{b}}, \hat{q}^{\mathcal{K}})]] = \text{var}[\mathbb{P}_n[\phi^{\mathcal{K}}(w^{\mathcal{K}}, \pi^{\mathrm{b}}, q^{\mathcal{K}})]] + o(n^{-1}h^{-1})$$

$$= \frac{1}{nh}\sum_{t=1}^{H} \mathbb{E}_{\tau_\theta} \left[ \frac{p_{\tau_\theta}(s_t)}{p_{\pi^{\mathrm{b}}}(s_t)\pi^{\mathrm{b}}(\tau_t(s_t) \mid s_t)}\text{var}[r_t + v_{t+1}^{\pi^{\mathrm{e}}}(s_{t+1}) \mid s_t, \tau_\theta(s_t)] \right] \Omega_2(k) + o(1/nh).$$

**Order of main constants in the bias and variance terms** Bias and variance terms are upper-bounded as follows.

**Bound of $B_H$**

$$\sum_{t=1}^{H} \left\{ \mathbb{E}_\tau \left[ \frac{r_t p^{(2)}(r_t|s_t, \tau_t(s_t))}{p(r_t|s_t, \tau_t(s_t))} \right] + \sum_{j=1}^{t-1} \mathbb{E}_\tau \left[ \frac{r_t p^{(2)}(s_{j+1} \mid s_j, \tau_j(s_j))}{p(s_{j+1} \mid s_j, \tau_j(s_j))} \right] \right\}$$

$$\leq R_{\max} \left\{ H G_1 + \frac{H(H-1)}{2} G_1 \right\}.$$

we used an argument:

$$|\mathbb{E}_\tau \left[ \frac{r_t p^{(2)}(r_t|s_t, \tau_t(s_t))}{p(r_t|s_t, \tau_t(s_t))} \right]| \leq \mathbb{E}_\tau \left[ \frac{r_t|p^{(2)}(r_t|s_t, \tau_t(s_t))|}{p(r_t|s_t, \tau_t(s_t))} \right] \leq R_{\max} \mathbb{E}_\tau \left[ \frac{|p^{(2)}(r_t|s_t, \tau_t(s_t))|}{p(r_t|s_t, \tau_t(s_t))} \right]$$

$$= R_{\max} \int |p^{(2)}(r_t|s_t, \tau_t(s_t))| p_\tau(s_t) \mathrm{d}(r_t, s_t)$$

$$\leq R_{\max} \| \int |p^{(2)}(r_t|s_t, \tau_t(s_t))| \mathrm{d}r_t \|_\infty \leq R_{\max} G_1^{(2)},$$

$$|\mathbb{E}_\tau \left[ \frac{r_t p^{(2)}(s_{j+1}|s_j, \tau_j(s_j))}{p(s_{j+1}|s_j, \tau_j(s_j))} \right]| \leq \mathbb{E}_\tau \left[ |\frac{r_t p^{(2)}(s_{j+1}|s_j, \tau_j(s_j))}{p(s_{j+1}|s_j, \tau_j(s_j))}| \right] \leq R_{\max} \mathbb{E}_\tau \left[ \frac{|p^{(2)}(s_{j+1}|s_j, \tau_j(s_j))|}{p(s_{j+1}|s_j, \tau_j(s_j))} \right]$$

$$= R_{\max} \int |p^{(2)}(s_{j+1}|s_j, \tau_j(s_j))| p_\tau(s_j) \mathrm{d}(s_{j+1}, s_j)$$

$$\leq R_{\max} \| \int |p^{(2)}(s_{j+1}|s_j, \tau_j(s_j))| p_\tau(s_j) \mathrm{d}s_{j+1} \|_\infty \leq R_{\max} G_2^{(2)}.$$

**Bound of $V_H$**

$$\sum_{t=1}^{H} \mathbb{E}_{\tau_\theta} \left[ \frac{p_{\tau_\theta}(s_t)}{p_{\pi^b}(s_t) \pi^b(\tau_\theta(s_t) \mid s_t)} \mathrm{var}[r_t + v_{t+1}^{\pi^e}(s_{t+1}) \mid s_t, \tau_\theta(s_t)] \right]$$

$$\leq \sum_{t=1}^{H} \mathbb{E}_{\tau_\theta} \left[ C_1 C_2 \mathrm{var}[r_t + v_{t+1}^{\pi^e}(s_{t+1}) \mid s_t, \tau_\theta(s_t)] \right]$$

$$= C_1 C_2 \mathrm{var}_{\tau_\theta} [\sum_{t=1}^{H} r_t] \leq C_1 C_2 R_{\max}^2 H^2.$$

### D.6 Proof of Theorem 5

**Replacing estimators with true functions** Here, we prove that nuisance estimators can be replace with true functions:

$$\mathbb{E}[A_n] = o((nh^3)^{-1/2}), \quad \mathrm{var}[A_n] = o(n^{-1}h^{-1}),$$

$$A_n = \mathbb{P}_{\mathcal{U}_1}[\psi^{\mathcal{K}}(\hat{w}^{\mathcal{K}[1]}, \hat{q}^{\mathcal{K}[1]}, \hat{d}^{w^{\mathcal{K}[1]}}, \hat{d}^{q^{\mathcal{K}[1]}}, \hat{\pi}^b) | \mathcal{U}_2] + \mathbb{P}_{\mathcal{U}_2}[\psi^{\mathcal{K}}(\hat{w}^{\mathcal{K}[2]}, \hat{q}^{\mathcal{K}[2]}, , \hat{d}^{w^{\mathcal{K}[2]}}, \hat{d}^{q^{\mathcal{K}[2]}}, \hat{\pi}^b) | \mathcal{U}_1]$$

$$- \mathbb{E}[\psi^{\mathcal{K}}(w^{\mathcal{K}}, q^{\mathcal{K}}, d^{w^{\mathcal{K}}}, d^{q^{\mathcal{K}}}, \pi^b)].$$

Then, what we have to prove is

$$\mathbb{E}[\mathbb{E}[\psi^{\mathcal{K}}(\hat{w}^{\mathcal{K}[1]}, \hat{q}^{\mathcal{K}[1]}, \hat{d}^{w^{\mathcal{K}[1]}}, \hat{d}^{q^{\mathcal{K}[1]}}, \hat{\pi}^{b[1]}) - \psi^{\mathcal{K}}(w^{\mathcal{K}}, q^{\mathcal{K}}, d^{w^{\mathcal{K}}}, d^{q^{\mathcal{K}}}, \pi^b) | \mathcal{U}_2]] = o((nh^3)^{-1/2}),$$

$$(32)$$

$$\mathbb{E}[\mathbb{E}[\{\psi^{\mathcal{K}}(\hat{w}^{\mathcal{K}[1]}, \hat{q}^{\mathcal{K}[1]}, \hat{d}^{w^{\mathcal{K}[1]}}, \hat{d}^{q^{\mathcal{K}[1]}}, \hat{\pi}^{b[1]}) - \psi^{\mathcal{K}}(w^{\mathcal{K}}, q^{\mathcal{K}}, d^{w^{\mathcal{K}}}, d^{q^{\mathcal{K}}}, \pi^b)\}^2 | \mathcal{U}_2]] = o((h)^{-1}).$$

$$(33)$$

From now on, we omit $[1], \mathcal{U}_2$.

Eq. (32) is proved by

$$\mathbb{E}[\mathbb{E}[\psi^{\mathcal{K}}(\hat{w}^{\mathcal{K}}, \hat{q}^{\mathcal{K}}, \hat{d}^{w^{\mathcal{K}}}, \hat{d}^{q^{\mathcal{K}}}, \hat{\pi}^b) - \psi^{\mathcal{K}}(w^{\mathcal{K}}, q^{\mathcal{K}}, d^{w^{\mathcal{K}}}, d^{q^{\mathcal{K}}}, \pi^b)|\mathcal{U}_1]]$$

$$= \mathbb{E}[\sum_{t=1}^{H} \{\hat{d}_t^{w^{\mathcal{K}}}(S_t)\hat{\eta}(S_t, A_t) - d_t^{w^{\mathcal{K}}}(S_t)\eta(S_t, A_t))\} K_h((A_t - \tau(S_t)))$$

$$\times (-\hat{q}_t^{\mathcal{K}}(S_t, A_t) + q_t^{\mathcal{K}}(S_t, A_t) + \hat{v}_{t+1}^{\mathcal{K}}(S_{t+1}) - v_{t+1}^{\mathcal{K}}(S_{t+1}))]$$

$$+ \mathbb{E}[\sum_{t=1}^{H} \left\{\hat{w}_t^{\mathcal{K}}(S_t)\hat{\eta}(S_t, A_t) - w_t^{\mathcal{K}}(S_t)\eta(S_t, A_t)\right\} K_h((A_t - \tau(S_t)))$$

$$\times (-d_t^{\hat{q}^{\mathcal{K}}}(S_t, A_t) + d_t^{q^{\mathcal{K}}}(S_t, A_t) + d_{t+1}^{\hat{v}^{\mathcal{K}}}(S_{t+1}) - d_{t+1}^{v^{\mathcal{K}}}(S_{t+1}))]$$

$$- \mathbb{E}[\sum_{t=1}^{H} \left\{\hat{w}_t^{\mathcal{K}}(S_t)\hat{\eta}(S_t, A_t) - w_t^{\mathcal{K}}(S_t)\eta(S_t, A_t)\right\} \times$$

$$h^{-2}k^{(1)}((A_t - \tau(S_t))h^{-1})\nabla\tau(S_t)(-\hat{q}_t^{\mathcal{K}}(S_t, A_t) + q_t^{\mathcal{K}}(S_t, A_t) + \hat{v}_{t+1}^{\mathcal{K}}(S_{t+1}) - v_{t+1}^{\mathcal{K}}(S_{t+1}))]$$

$$= o(h^{-2}) + \mathbb{E}[\sum_{t=1}^{H} \max\{\|\hat{w}_t^{\pi^e} - w_t^{\pi^e}\|_\infty, \|\hat{\eta}_t - \eta_t\|_{1,\infty}, \|\hat{d}_t^{w^{\mathcal{K}}} - d_t^{w^{\mathcal{K}}}\|_\infty\}$$

$$\times \max\{\|\hat{q}_t - q_t\|_{1,\infty}, \|\hat{q}_{t+1} - q_{t+1}\|_{1,\infty}, \|\hat{d}_t^{q^{\mathcal{K}}} - d_t^{q^{\mathcal{K}}}\|_\infty, \|\hat{d}_{t+1}^{q^{\mathcal{K}}} - d_{t+1}^{q^{\mathcal{K}}}\|_\infty\}]$$

$$= o(n^{-1/2}h^{-3/2}).$$

Eq. (33) is similarly proved.

**Bias part** First, we have

$$\mathbb{E}[\mathbb{P}_n[\psi(w^{\mathcal{K}}, q^{\mathcal{K}}, d^{w^{\mathcal{K}}}, d^{q^{\mathcal{K}}}, \pi^b)]] = \nabla\mathbb{E}[\sum_t \lambda_t^{\mathcal{K}} r_t] - \nabla J(\theta) = \nabla\{\sum_t \mathbb{E}[\{\lambda_t^{\mathcal{K}} - \lambda_t\} r_t]\}.$$

Here, we use a doubly robust property of $\phi$. Then, the above is equal to

$$-\sum_t \left\{\int c(s_t, a_t) \sum_{i=1}^{t} \frac{k^{(1)}((a_i - \tau(s_i))h^{-1})}{h^2}\nabla\tau_i(s_i) \prod_{j\neq i}^{t} \frac{k((a_j - \tau(s_j))h^{-1})}{h} \prod_{l=1}^{t} p(s_l \mid s_{l-1}, a_{l-1})\}d(h_{a_t}) - \mathbb{E}[\lambda_t r_t]\right\}$$

$$= -\sum_{t=1}^{H} \left\{\int c(s_t, \tau_t(s_t) + hu_t) \sum_{i=1}^{t} h^{-1}k^{(1)}(u_i)\nabla\tau_i(s_i) \prod_{j\neq i}^{t} k(u_j) \prod_{l=1}^{t} p(s_l \mid s_{l-1}, \tau(s_{l-1}) + hu_l)d(h_{a_t}^u) - \mathbb{E}[\lambda_t r_t]\right\}$$

$$= \sum_{t=1}^{H} \{\int c_t^{(1)}(s_t, \tau_t(s_t) + hu_t)\nabla\tau_t(s_t) \prod_{j=1}^{t} k(u_j) \prod_{i=1}^{t} p(s_l \mid s_{i-1}, \tau(s_{i-1}) + hu_i)d(h_{a_t}^u) +$$

$$+ \int c(s_t, \tau_t(s_t) + hu_t) \sum_{i=1}^{t} \nabla\tau_i(s_i)\frac{p^{(1)}(s_i \mid s_{i-1}, \tau(s_{i-1}) + hu_i)}{p(s_i \mid s_{i-1}, \tau(s_{i-1}) + hu_i)} \prod_{j=1}^{t} k(u_j) \prod_{l=1}^{t} p(s_l \mid s_{l-1}, \tau(s_{l-1}) + hu_l)d(h_{a_t}^u)$$

$$- \mathbb{E}[\lambda_t r_t]\}.$$

This is equal to

$$0.5h^2 M_2(\mathcal{K}) \sum_t \{\int c_t^{(3)}(s_t, \tau_t(s_t)) \prod_{i=1}^{t} p(s_l \mid s_{i-1}, \tau(s_{i-1})) \mathrm{d}(h_{s_t})+$$

$$\int c_t^{(1)}(s_t, \tau_t(s_t)) \nabla \tau_t(s_t) \sum_{j=2}^{t} \frac{p^{(2)}(s_j \mid s_{j-1}, \tau(s_{j-1}))}{p(s_l \mid s_{i-1}, \tau(s_{i-1}))} \prod_{l=1}^{t} p(s_l \mid s_{i-1}, \tau(s_{i-1})) \mathrm{d}(h_{s_t}^{-u})+$$

$$+ \int c_t^{(2)}(s_t, \tau_t(s_t)) \sum_{i=2}^{t} \nabla \tau_i(s_i) \frac{p^{(1)}(s_i \mid s_{i-1}, \tau(s_{i-1}))}{p(s_i \mid s_{i-1}, \tau(s_{i-1}))} \prod_{l=1}^{t} p(s_l \mid s_{l-1}, \tau(s_{l-1})) \mathrm{d}(h_{s_t}^{-u})$$

$$+ \int c(s_t, \tau_t(s_t)) \sum_{i=2}^{t} \nabla \tau_i(s_i) \left\{ \frac{p^{(3)}(s_i \mid s_{i-1}, \tau(s_{i-1}))}{p(s_i \mid s_{i-1}, \tau(s_{i-1}))} + \frac{p^{(1)}(s_i \mid s_{i-1}, \tau(s_{i-1}))}{p(s_i \mid s_{i-1}, \tau(s_{i-1}))} \sum_{j \neq i} \frac{p^{(2)}(s_j \mid s_{j-1}, \tau(s_{j-1}))}{p(s_j \mid s_{j-1}, \tau(s_{j-1}))} \right\}$$

$$\times \prod_{l=1}^{t} p(s_l \mid s_{l-1}, \tau(s_{l-1})) \mathrm{d}(h_{s_t}^{-u})$$

$$= 0.5h^2 M_2(k) \tilde{B}_H.$$

Here, $h_{a_t}^{-u} = \{s_1, s_2, \cdots\}$. In the end, $\tilde{B}_H$ is equal to

$$\sum_{t=1}^{H} \left\{ \mathbb{E}_\tau \left[ r_t \frac{p^{(3)}(r_t \mid s_t, \tau_t(s_t))}{p(r_t \mid s_t, \tau_t(s_t))} \nabla \tau_t(s_t) \right] + \sum_{j=1}^{t-1} \mathbb{E}_\tau \left[ r_t \frac{p^{(2)}(r_t \mid s_t, \tau_t(s_t)) p^{(1)}(s_{j+1} \mid s_j, \tau_j(s_j))}{p(r_t \mid s_t, \tau_t(s_t)) p(s_{j+1} \mid s_j, \tau_j(s_j))} \nabla \tau_j(s_j) \right] \right\} +$$

$$+ \sum_{t=1}^{H} \sum_{j=1}^{t-1} \mathbb{E}_\tau \left[ r_t \frac{p^{(1)}(r_t \mid s_t, \tau_t(s_t)) p^{(2)}(s_{j+1} \mid s_j, \tau_j(s_j))}{p(r_t \mid s_t, \tau_t(s_t)) p(s_{j+1} \mid s_j, \tau_j(s_j))} \nabla \tau_t(s_t) \right] +$$

$$+ \sum_{t=1}^{H} \left\{ \sum_{j=1}^{t-1} \mathbb{E}_\tau \left[ r_t \frac{p^{(3)}(s_{j+1} \mid s_j, \tau_j(s_j))}{p(s_{j+1} \mid s_j, \tau_j(s_j))} \nabla \tau_j(s_j) \right] + \sum_{j \neq i}^{t-1} \mathbb{E}_\tau \left[ r_t \frac{p^{(1)}(s_{i+1} \mid s_i, \tau_i(s_i)) p^{(2)}(s_{j+1} \mid s_j, \tau_j(s_j))}{p(s_{i+1} \mid s_i, \tau_i(s_i)) p(s_{j+1} \mid s_j, \tau_j(s_j))} \nabla \tau_i(s_i) \right] \right\}.$$

The operator norm of $\otimes \tilde{B}_H$ is upper bounded by

$$R_{\max}^2 \Upsilon^2 \left\{ H G_1^{(3)} + \frac{H(H-1)}{2} \{G_1^{(2)} G_2^{(1)} + G_1^{(1)} G_2^{(2)} + G_2^{(3)}\} + \frac{H(H-1)(H-2)}{3} G_2^{(1)} G_2^{(2)} \right\}^2.$$

For example,

$$\left\| \otimes \mathbb{E}_\tau \left[ r_t \frac{p^{(3)}(s_{j+1} \mid s_j, \tau_j(s_j))}{p(s_{j+1} \mid s_j, \tau_j(s_j))} \nabla \tau_j(s_j) \right] \right\|_{\mathrm{op}} \leq \left\| \otimes \mathbb{E}_\tau \left[ \frac{|r_t p^{(3)}(s_{j+1} \mid s_j, \tau_j(s_j))|}{p(s_{j+1} \mid s_j, \tau_j(s_j))} \nabla \tau_j(s_j) \right] \right\|_{\mathrm{op}}$$

$$\leq R_{\max}^2 G_2^{2(3)} \left\| \otimes \mathbb{E}_\tau \left[ \nabla \tau_j(s_j) \right] \right\| \|_{\mathrm{op}}$$

$$\leq R_{\max}^2 G_2^{2(3)} \left\| \mathbb{E}_\tau \left[ \otimes \nabla \tau_j(s_j) \right] \right\|_{\mathrm{op}} \leq R_{\max}^2 G_2^{2(3)} \Upsilon^2.$$

**Variance part**   The variance part is calculated as

$$\frac{1}{n} \sum_{h=1}^{H} \mathbb{E} \left[ \frac{w_t^{2\mathcal{K}}(S_t) k^{2(1)}(\{(A_t - \tau(S_t)\} h^{-1})}{h^4 \pi_t^{2b}(A_t \mid S_t)} \mathrm{var}[R_t + v_{t+1}^\mathcal{K}(S_{t+1}) \mid S_t, A_t] \otimes \nabla \tau(S_t) \right] + o(h^3/n)$$

$$= \frac{1}{nh^3} \sum_{h=1}^{H} \int k^{2(1)}(u_h) \mathrm{d}(u_h) \mathbb{E} \left[ \frac{w^{2\mathcal{K}}(S_t)}{\pi_t^{2b}(\tau(S_t) \mid S_t)} \mathrm{var}[R_t + v_{t+1}^{\pi^e}(S_{t+1}) \mid S_t, \tau(S_t)] \otimes \nabla \tau(S_t) \right] + o(h^3/n)$$

$$= \frac{\Omega_2^{(1)}(k)}{nh^3} \sum_{h=1}^{H} \mathbb{E}_\tau \left[ \frac{w^{\pi^e}(s_t)}{\pi_t^b(\tau_t(s_t) \mid s_t)} \mathrm{var}[r_t + v_{t+1}^{\pi^e}(s_{t+1}) \mid s_t, \tau_t(s_t)] \otimes \nabla \tau_t(s_t) \right] + o(h^3/n)$$

$$= \frac{\Omega_2^{(1)}(k)}{nh^3} \tilde{V}_H + o(h^3/n).$$

Then, the operator norm of $\tilde{V}_H$ is upper bounded as

$$\left\| \sum_{t=1}^{H} \mathbb{E}_\tau \left[ \frac{w^{\pi^{\mathrm{e}}}(s_t)}{\pi_t^b(\tau_t(s_t)|s_t)} \mathrm{var}[r_t + v_{t+1}^{\pi^{\mathrm{e}}}(s_{t+1})|s_t, \tau_t(s_t)] \otimes \nabla \tau_t(s_t) \right] \right\|_{\mathrm{op}}$$

$$\leq \left\| C_1 C_2 \sum_{t=1}^{H} \mathbb{E}_\tau \left[ \mathrm{var}[r_t + v_{t+1}^{\pi^{\mathrm{e}}}(s_{t+1})|s_t, \tau_t(s_t)] \otimes \nabla \tau_t(s_t) \right] \right\|_{\mathrm{op}}$$

$$\leq \left\| C_1 C_2 \| \otimes \nabla \tau(s) \|_\infty \sum_{t=1}^{H} \mathbb{E}_\tau \left[ \mathrm{var}[r_t + v_{t+1}^{\pi^{\mathrm{e}}}(s_{t+1})|s_t, \tau_t(s_t)] \right] \right\|_{\mathrm{op}}$$

$$\leq \| C_1 C_2 \| \otimes \nabla \tau(s) \|_\infty R_{\max}^2 H^2 ] \|_{\mathrm{op}} \leq C_1 C_2 R_{\max} H^2 \Upsilon^2.$$

## E    Different Representation of Theorem 1

**Theorem 7.** *Suppose for $i = 1, 2$, $\|\hat{\pi}^{b,[i]}(A \mid S) - \pi^b(A \mid S)\|_{1,\infty} = \mathrm{o}_p(1)$, $\|\hat{q}^{[i]}(S,A) - q(S,A)\|_{1,\infty} = \mathrm{o}_p(1)$, $\|\hat{\pi}^{b,[i]}(A \mid S) - \pi^b(A \mid S)\|_\infty \|\hat{q}^{[i]}(S,A) - q(S,A)\|_\infty = \mathrm{o}_p((nh)^{-1/2})$, $nh^5 = \mathcal{O}(1)$, $nh \to \infty$, that $\pi^{\mathrm{b}}(a \mid s)$, $q(s,a)$ are twice continuously differentiable wrt $a$ for almost all $s$, and that $\hat{\pi}^{b,[i]}, \hat{q}^{[i]}$ are uniformly bounded by a constant. Then, there exists $F_n$ s.t. $\hat{J}^{\mathcal{D}} - J = \mathbb{E}_n \left[ \frac{K_h(A-\tau(S))\{R-f_1\}}{f_2} + f_3 \right] - J + \mathrm{o}_p((nh)^{-1/2})$, where*

$$\mathbb{E}[\mathbb{E}_n \left[ \frac{K_h(A - \tau(S))\{R - f_1\}}{f_2} + f_3 \right] - J] = B + \mathrm{o}((nh)^{-1/2}),$$

$$\mathrm{var}[\mathbb{E}_n \left[ \frac{K_h(A - \tau(S))\{R - f_1\}}{f_2} + f_3 \right]] = V + \mathrm{o}((nh)^{-1}),$$

$$B = \mathbb{E}[q^{(2)}(S, \tau(S)) + 2q^{(1)}(S, \tau(S))\pi^{b(1)}(\tau(S)|S)/\pi^b(\tau(S)|S)], \quad V = \frac{1}{nh}\mathbb{E} \left[ \frac{\mathrm{var}[R \mid S, \tau(S)]}{\pi^b(\tau(S) \mid S)} \right].$$

*If additionally $\hat{\pi}^{b,[i]}(a \mid s)$, $\hat{q}^{[i]}(s,a)$ are twice continuously differentiable wrt $a$, then the same holds for $\hat{J}^{\mathcal{K}}$ with $B = \mathbb{E}[q^{(2)}(S, \tau(S))]$ and the same $V$ as the above.*

We prove Theorem 7. To do that, we prove Theorem 8. The rest of the proof is the same the that of Theorem 1.

**Theorem 8.** *In the case $\mathcal{K}$, assuming (b), we have*

$$\hat{J} = \mathbb{E}_n \left[ \frac{K_h(A-\tau(S))\{R-f_1\}}{f_2} + f_3 \right] + \mathrm{o}_p((nh)^{-1/2}). \tag{34}$$

*In the case $\mathcal{D}$, the above (34) also holds assuming (a).*

### E.0.1    Proof of Theorem 8, Case $\mathcal{K}$

Let us define

$$\phi_1(s, a, r; q, \pi^{\mathrm{b}}) = \frac{K_h(a-\tau(s))\{r-q(s,a)\}}{\pi^{\mathrm{b}}(a|s)} + v(s), v(s) = \int K_h(a - \tau(s))q(s,a)\mathrm{d}(a).$$

As in a standard argument (Chernozhukov et al., 2018; Kallus and Uehara, 2020a), what we have to prove is

$$\mathbb{E}[\phi_1(S, A, R; \hat{q}^{[1]}, \hat{\pi}^{b[1]}) - \phi_1(S, A, R; q, \pi^b) \mid \mathcal{U}_2] = \mathrm{o}_p((nh)^{-1/2}), \tag{35}$$

$$\mathbb{E}[\{\phi_1(S, A, R; \hat{q}^{[1]}, \hat{\pi}^{b[1]}) - \phi_1(S, A, R; q, \pi^b)\}^2 \mid \mathcal{U}_2] = \mathrm{o}_p(h^{-1}). \tag{36}$$

Then, the same argument holds for $\phi_1(S, A, R; \hat{q}^{[2]}, \hat{\pi}^{b[2]}) - \phi_1(S, A, R; q, \pi^b)$. The desired statement is concluded since

$$\mathbb{P}_n[\phi_1(S, A, R; \hat{q}, \hat{\pi}^b)] - \mathbb{P}_n[\phi_1(S, A, R; q, \pi^b)]$$
$$= \mathbb{G}_n[\phi_1(S, A, R; \hat{q}, \hat{\pi}^b) - \phi_1(S, A, R; q, \pi^b) \mid \mathcal{U}_2] \tag{37}$$
$$+ \mathbb{E}[\phi_1(S, A, R; \hat{q}, \hat{\pi}^b) - \phi_1(S, A, R; q, \pi^b) \mid \mathcal{U}_2]$$
$$= \mathrm{o}_p((nh)^{-1/2}) + \mathrm{o}_p((nh)^{-1/2}).$$

Especially, (37) is $o_p((nh)^{-1/2})$ since for $\epsilon > 0$,

$$\mathbb{P}(n^{1/2}h^{1/2} \times \mathbb{G}_{\mathcal{U}_2}[\phi_1(S, A, R; \hat{q}^{[1]}, \hat{\pi}^{b[1]}) - \phi_1(S, A, R; q, \pi^b)] > \epsilon|\mathcal{U}_2)$$
$$\leq \mathbb{E}[h\{\phi_1(S, A, R; \hat{q}^{[1]}, \hat{\pi}^{b[1]}) - \phi_1(S, A, R; q, \pi^b)\}^2/\epsilon^2|\mathcal{U}_2] = o_p(1).$$

Therefore, noting $\mathbb{P}(n^{1/2}h^{1/2}\mathbb{G}_{\mathcal{U}_2}[\phi_1(S, A, R; \hat{q}^{[1]}, \hat{\pi}^{b[1]}) - \phi_1(S, A, R; q, \pi^b)] > \epsilon|\mathcal{U}_2)$ is uniformly integrable,

$$\mathbb{P}(n^{1/2}h^{1/2}\mathbb{G}_{\mathcal{U}_2}[\phi_1(S, A, R; \hat{q}^{[1]}, \hat{\pi}^{b[1]}) - \phi_1(S, A, R; q, \pi^b)] > \epsilon|\mathcal{U}_2) = o_p(1)$$

implies

$$\mathbb{G}_{\mathcal{U}_2}[\phi_1(S, A, R; \hat{q}^{[1]}, \hat{\pi}^{b[1]}) - \phi_1(S, A, R; q, \pi^b)] = o_p(n^{-1/2}h^{-1/2}).$$

**Proof of Eq. (35) and Eq. (36)** In this subsection, we remove $\{[1]\}$ for the ease of the notation. To prove (9), we show

$$\mathbb{E}\left[K_h(A - \tau(S))\left\{\frac{1}{\hat{\pi}^b(A|S)} - \frac{1}{\pi^b(A|S)}\right\}\{q(S, A) - \hat{q}(S, A)\} \mid \mathcal{U}_2\right] = o_p((nh)^{-1/2}).$$

This is proved by

$$\mathbb{E}\left[K_h(A - \tau(S))\left\{\frac{1}{\hat{\pi}^b(A|S)} - \frac{1}{\pi^b(A|S)}\right\}\{q(S, A) - \hat{q}(S, A)\} \mid \mathcal{U}_2\right]$$

$$= \int \frac{1}{h}k((a - \tau(s))h^{-1})\left\{\frac{1}{\hat{\pi}^b(a|s)} - \frac{1}{\pi^b(a|s)}\right\}\{q(s, a) - \hat{q}(s, a)\}\pi^b(a \mid s)p(s)\mathrm{d}(s, a)$$

$$= \int k(u)\left\{\frac{1}{\hat{\pi}^b(\tau(s) + uh|s)} - \frac{1}{\pi^b(\tau(s) + uh|s)}\right\}\{q(s, \tau(s) + uh) - \hat{q}(s, \tau(s) + uh)\}\pi^b(\tau(s) + uh \mid s)p(s)\mathrm{d}(u, s)$$

$$= \int k(u)\left\{\frac{1}{\hat{\pi}^b(\tau(s)|s)} - \frac{1}{\pi^b(\tau(s)|s)} + uh\left\{-\frac{\hat{\pi}^{b(1)}(\tau(s)|s)}{\hat{\pi}^{2b}(\tau(s)|s)} + \frac{\pi^{b(1)}(\tau(s)|s)}{\pi^{2b}(\tau(s)|s)}\right\} + \mathcal{O}((uh^2))\right\} \times$$

$$\{q(s, \tau(s)) - \hat{q}(s, \tau(s)) + uh\{q^{(1)}(s, \tau(s)) - \hat{q}^{(1)}(s, \tau(s))\} + \mathcal{O}((uh^2))\} \times$$

$$\{\pi^b(\tau(s) \mid s) + uh\pi^{b(1)}(\tau(s) \mid s) + \mathcal{O}((uh^2))\}p(s)\mathrm{d}(u, s)$$

$$= M_2(k)\int\left\{\frac{1}{\hat{\pi}^b(\tau(s) \mid s)} - \frac{1}{\pi^b(\tau(s) \mid s)}\right\}\{q(s, \tau(s)) - \hat{q}(s, \tau(s))\}\pi^b(\tau(s) \mid s)p(s)\mathrm{d}(s) + \mathcal{O}(h^2) \times o_p(1)$$

$$= o_p((nh)^{-1/2}).$$

In the last line, we use the assumptions that $q(a, x), \pi^b(a|x), \hat{q}(a, x), \hat{\pi}^b(a|x)$ are $C^2$-functions wrt actions, and

$$\left\|\frac{1}{\hat{\pi}^b(A \mid S)} - \frac{1}{\pi^b(A \mid S)}\right\|_\infty \|q(S, A) - \hat{q}(S, A)\|_\infty = o_p(n^{-1/2}h^{-1/2}),$$

$$\left\|\frac{1}{\hat{\pi}^b(A) \mid S)} - \frac{1}{\pi^b(A \mid S)}\right\|_{1,\infty} = o_p(1), \|\hat{q}(S, A) - \hat{q}(S, A)\|_{1,\infty} = o_p(1),$$

$$\mathcal{O}(h^2) \times o_p(1) = o_p((nh)^{-1/2}), nh^5 = \mathcal{O}(1).$$

In addition, Eq. (36) is proved since

$$\mathbb{E}[\{\phi_1(S, A, R; \hat{q}^{[1]}, \hat{\pi}^{b[1]}) - \phi_1(S, A, R; q, \pi^b)\}^2 \mid \mathcal{U}_2]$$

$$\lesssim \mathbb{E}\left[K_h(A - \tau(S))^2\left\{\frac{1}{\hat{\pi}^b(A \mid S)} - \frac{1}{\pi^b(A \mid S)}\right\}^2\{q(S, A) - \hat{q}(S, A)\}^2 \mid \mathcal{U}_2\right]$$

$$+ \mathbb{E}\left[K_h(A - \tau(S))^2\left\{\frac{1}{\hat{\pi}^b(A \mid S)} - \frac{1}{\pi^b(A \mid S)}\right\}^2\{R - q(S, A)\}^2 \mid \mathcal{U}_2\right]$$

$$+ \mathbb{E}\left[\frac{K_h(A - \tau(S))^2}{\pi^b(A \mid S)^2}\{q(S, A) - \hat{q}(S, A)\}^2 \mid \mathcal{U}_2\right] + \mathbb{E}\left[\{\hat{v}(S) - v(S)\}^2 \mid \mathcal{U}_2\right]$$

$$\lesssim h^{-1}\max\left\{\|\hat{q}(S, \tau(S)) - q(S, \tau(S))\|_2^2, \left\|\frac{1}{\hat{\pi}^b(\tau(S)|S)} - \frac{1}{\pi^b(\tau(S)|S)}\right\|_2^2\right\} + \mathcal{O}(1) = o_p(h^{-1}).$$

### E.0.2 Proof of Theorem 8, Case $\mathcal{D}$

Essentially, the same proof is seen in Colangelo and Lee (2019). For completeness, we also write the proof here with our notation. Let us define

$$\phi_2(s,a,r;q,\pi^{\mathrm{b}}) = \frac{K_h(a-\tau(s))\{r-q(s,\tau(s))\}}{\pi^{\mathrm{b}}(a|s)} + q(S,\tau(S)).$$

As in a standard argument similar to the case $\mathcal{K}$, what we have to prove is

$$\mathbb{E}[\phi_2(S,A,R;\hat{q}^{[1]},\hat{\pi}^{b[1]}) - \phi_2(S,A,R;q,\pi^b)|\mathcal{U}_2] = \mathrm{o}_p((nh)^{-1/2}), \tag{38}$$

$$\mathbb{E}[\{\phi_2(S,A,R;\hat{q}^{[1]},\hat{\pi}^{b[1]}) - \phi_2(S,A,R;q,\pi^b)\}^2|\mathcal{U}_2] = \mathrm{o}_p(h^{-1}). \tag{39}$$

In this subsection, we remove $\{[1]\}$ for the ease of the notation.

Eq. (35) is proved since

$$\mathbb{E}[\phi_2(S,A,R;\hat{q},\hat{\pi}^b) - \phi_2(S,A,R;q,\pi^b) \mid \mathcal{U}_2]$$

$$= \mathbb{E}\left[K_h(A-\tau(S))\left\{\frac{1}{\hat{\pi}^b(\tau(S)\mid S)} - \frac{1}{\pi^b(\tau(S)\mid S)}\right\}\{q(S,\tau(S)) - \hat{q}(S,\tau(S))\} \mid \mathcal{U}_2\right] + \tag{40}$$

$$+ \mathbb{E}\left[K_h(A-\tau(S))\left\{\frac{1}{\hat{\pi}^b(\tau(S)\mid S)} - \frac{1}{\pi^b(\tau(S)\mid S)}\right\}\{R-q(S,\tau(S))\} \mid \mathcal{U}_2\right] \tag{41}$$

$$+ \mathbb{E}\left[\frac{K_h(A-\tau(S))}{\pi^b(\tau(S)\mid S)}\{q(S,\tau(S)) - \hat{q}(S,\tau(S))\} + \hat{q}(S,\tau(S)) - q(S,\tau(S)) \mid \mathcal{U}_2\right] \tag{42}$$

$$= \mathrm{o}_p((nh)^{-1/2}) + \mathrm{o}_p(1) \times \mathcal{O}(h^2) + \mathrm{o}_p(1) \times \mathcal{O}(h^2) = \mathrm{o}_p(nh)^{-1/2}.$$

Here, we use the facts that (40) is $\mathrm{o}_p((nh)^{-1/2})$, (41) is $\mathrm{o}_p(1) \times \mathcal{O}(h^2)$, (42) is $\mathrm{o}_p(1) \times \mathcal{O}(h^2)$, which we will prove soon. In the last line, we use $nn^5 = \mathcal{O}(1)$. From now on, we prove (41) is $\mathrm{o}_p(1) \times \mathcal{O}(h^2)$:

$$\mathbb{E}\left[K_h(A-\tau(S))\left\{\frac{1}{\hat{\pi}^b(\tau(S)\mid S)} - \frac{1}{\pi^b(\tau(S)\mid S)}\right\}\{R-q(S,\tau(S))\} \mid \mathcal{U}_2\right]$$

$$= \mathbb{E}\left[\left\{\frac{1}{\hat{\pi}^b(\tau(S)\mid S)} - \frac{1}{\pi^b(\tau(S)\mid S)}\right\}\{\mathbb{E}[K_h(A-\tau(S))q(S,A) \mid S] - K_h(A-\tau(S))q(S,\tau(S))\} \mid \mathcal{U}_2\right]$$

$$= \mathbb{E}\left[\left\{\frac{1}{\hat{\pi}^b(\tau(S)\mid S)} - \frac{1}{\pi^b(\tau(S)\mid S)}\right\}\{\mathcal{O}(h^2)\} \mid \mathcal{U}_2\right]$$

$$= \mathrm{o}_p(1) \times \mathcal{O}(h^2).$$

More specifically,

$$\mathbb{E}[K_h(A-\tau(S))\{q(S,A) - q(S,\tau(S))\} \mid S] = \int \frac{1}{h}k\left\{\frac{a-\tau(s)}{h}\right\}\pi^{\mathrm{b}}(a|s)\{q(s,a) - q(s,\tau(s))\}\mathrm{d}a$$

$$= \int k(u)\pi^{\mathrm{b}}(\tau(s)+uh|s)\{q(s,\tau(s)+uh) - q(s,\tau(s))\}\mathrm{d}u$$

$$= \int k(u)\{\pi^{\mathrm{b}}(\tau(s)|s) + \mathcal{O}(uh)\}\{uhq^{(1)}(s,\tau(s)) + \mathcal{O}(h^2)\}\mathrm{d}u = \mathcal{O}(h^2).$$

noting $\int uk(u)\mathrm{d}u = 0$. Next, we prove (15) is $\mathrm{o}_p(1) \times \mathcal{O}(h^2)$:

$$\mathbb{E}\left[\frac{K_h(A-\tau(S))}{\pi^b(\tau(S)\mid S)}\{q(S,\tau(S)) - \hat{q}(S,\tau(S))\} + \hat{q}(S,\tau(S)) - q(S,\tau(S)) \mid \mathcal{U}_2\right]$$

$$= \mathbb{E}\left[\left\{\frac{K_h(A-\tau(S))}{\pi^b(\tau(S)\mid S)} - 1\right\}\{q(S,\tau(S)) - \hat{q}(S,\tau(S))\} \mid \mathcal{U}_2\right]$$

$$= \mathbb{E}\left[\{\mathcal{O}(h^2)\}\{q(S,\tau(S)) - \hat{q}(S,\tau(S))\} \mid \mathcal{U}_2\right] = \mathrm{o}_p(1) \times \mathcal{O}(h^2).$$

Eq. (36) is similarly proved as in the case $\mathcal{K}$.