[Reviews · NeurIPS 2020]

Review 1

Summary and Contributions: The paper investigates the problem of offline reinforcement learning when the behaviour policy is deterministic. The paper provides an asymptotic MSE analysis and importantly show that the rate is independent of the horizon length, a common issue in OPE which is known as the curse of the horizon. ======================== After reading the rebuttal, I think that the authors have made a good effort in responding to the concerns and have agreed to add more experiments and explanations. I am looking forward to see these additions and keep my score as is.

Strengths: The paper tackles an important limitation of previous work: the assumption that the IS ratio exist. In the case of deterministic policies it is not well defined, therefore this is an important contribution. To go around this assumption the authors employ a stochastic relaxation (based on parameter h) of the deterministic policy and propose different combinations to approximate to kernerlize the deterministic policy. The analysis is done on two of these combinations in the policy evaluation and policy gradient settings, which covers most applications .

Weaknesses: In the introduction it is said that “applying these methods in a straightforward manner to RL may lead to a bad convergence rate that deteriorates with horizon.” L31-32. It would make a more solid argument to have more of a discussion on this with respect to the presented results. Something is said in Remark 2, i.e. [1] assumes a known behaviour policy, but it would be interesting to know more about what that entails to. Moreover, in the experiments, the behaviour policy is also assumed to be known. Is there a reason for doing this? What would be the results when it is unknown? Out of the 9 combinations, only two are investigated. Is there a particular reason for choosing these two? In the beginning of section 4, reward and next state densities are assumed thrice differentiable. These seems like an assumption hard to respect in practice. Do the authors see a way around this? Although half of the paper is concerned with the policy evaluation setting, all the experiments are done in the policy gradient setting. This seems like a curious choice and the paper would be more complete and compelling with additional policy evaluation experiments. Moreover, the experiments are executed for a fixed value of H. It would be particularly interesting to know how the cumulative algorithms versus marginal ones compare, especially given recent papers suggesting that one is not necessarily always better than the other [2].

Correctness: The claims and methodology seem correct.

Clarity: The paper is well written and situates the contributions through relevant previous publications. In general, it would be interesting to know if the derivation itself is very different/more complicated than previous work. [3]

Relation to Prior Work: The prior work is discussed.

Reproducibility: Yes

Additional Feedback: Typos: “CDGD and CDGK” L290 “MPRK MPKD” Figure 1 “the former is known to suffer from the high variance and the latter from model misspecification.” L72-73 -> isn’t the inverse? [1] Bibaut and van der Laan Data-adaptive smoothing for optimal-rate estimation of possibly non-regular parameters [2] Liu, Bacon and Brunskill, Understanding the Curse of Horizon in Off-Policy Evaluation via Conditional Importance Sampling [3] Kallus and Uehara, Double reinforcement learning for efficient off-policy evaluation in markov decision processes


Review 2

Summary and Contributions: [After rebuttal: Thank you. I have read your rebuttal and I am happy with it (especially in explaining the existence of density ratio). I would also like to see a short description of derivation steps, a better presentation of their notations and additional experiment results with different hyperparameter setups added in the final paper because I think that would make the paper more accessible to the NeurIPS readers who are interested in learning the techniques from the paper. Overall, I keep my score as it is.] ======== This paper addresses the problem of estimating off-policy value and gradient of deterministic polices with continuous actions by proposing general doubly robust estimators. This is done in general by (i) kernelizing the discrepancy between the observed action and the deterministic policy and (ii) allowing general nuisance functions whose specific realizations cover several previous estimators such as Kallus and Zhou (2018). The paper covers both the bandit setting (H=1) and extends to RL setting (H > 1). The paper then carefully establishes the bias, variance and convergence rate of their estimators.

Strengths: -The results are comprehensive: it covers both value and gradient estimation in both bandit and RL settings with rigorous analysis of their proposed estimators. -New interesting results: In the bandit setting for policy evaluation, they have achieved an improved variance as compared to Kallus and Zhou (2018) while in the RL setting for policy evaluation, their proposed estimators (MDRD, MDRK) do not suffer from the curse of horizon.

Weaknesses: -The experimental setting is relatively simple where the hyper parameters seem arbitrary. For example, why choosing N(0.0.3^2) and N(0.8s, 1.0^2)? How about the results for different hyper-parameters or models? There is only one experiment thus it is difficult to conclude if the proposed estimators, though very theoretically sound, are reliable in practice. -The paper content is quite dense to follow (see more in the Clarity section)

Correctness: The method and the empirical methodology are correct. The theoretical claims appear correct (see Additional Feedback and comments section).

Clarity: The overall structure are well planned and the paper is well written in general. However, I find it generally hard to follow the paper because (i) the paper content is quite dense with too many notations to keep up with, and (ii) some derivation steps are non-trivial and are not well explained in the paper (e.g., in proving Theorem 1 in line 497, I assume Taylor’s theorem is applied there and it would be nice to explicitly explain so in the proofs).

Relation to Prior Work: The paper carefully discuss previous related works.

Reproducibility: Yes

Additional Feedback: - Line 72-73: “The former [direct method] is known to suffer from the high variance”. It is a bit confusing here because DM is known to have low variance (but nonzero bias) (e.g., see Farajtabar et al. “More Robust Doubly Robust Off-policy Evaluation”). Is this a typo here? - Line 81-82: I do not fully understand the reason of nonexistence of density ratio stated at these lines. Why for the density ratio to exist in the case of deterministic \pi^e it requires \pi^b_t(a_t|s_t) to have atom at \tau_t(s_t)? As I understand, for the density ratio \pi^e_t(a_t|s_t) / \pi^b_t(a_t|s_t) to exist, it requires that \pi^e is absolutely continuous w.r.t. \pi^b. In the case of deterministic policy \pi^e_t(s_t) = \tau_t(s_t), this condition translates into that that \pi^b_t(\tau_t(s_t) | s_t) is nonzero which is the case for stochastic behaviour policy. In addition, kernelized estimator in Eq. (2) is used to smooth the hard rejection in IS because P( \tau_t(s_t) = a_t ) = 0 in continuous setting. So, to me it appears that the density ratio still exists in the case of deterministic policies and continuous actions; it is just that the importance sampling cannot apply because it would reject all the samples. But I might be wrong here and happy to be corrected in this case.


Review 3

Summary and Contributions: This paper proposed doubly robust off policy value and gradient estimation for deterministic policies. Since the density ratio does not exist for deterministic policies, the author proposed to approximate the deterministic policy by kernel smoothing. Based on the idea, the author studied and analyzed the asymptotic mean square error of several doubly robust gradient estimators under different settings, including contextual bandit and RL.

Strengths: The paper is well-written and the core idea is easy to follow. The analysis is novel and easy to follow.

Weaknesses: 1.The novelty of the proposed idea is limited. It seems that the proposed estimator is just to incorporate the kernel smoothing idea from [1] to the existing doubly robust estimators by replacing the original IS estimators. 2. The evaluation is too simple. I am curious the performance when the dimension of the action is high and empirical performance on offline benchmarks[2].

Correctness: Yes

Clarity: Yes

Relation to Prior Work: Yes

Reproducibility: No

Additional Feedback: Since the proposed method can directly applied to offline RL tasks, the author should have some initial or empirical results on control tasks or medical treatments. And also, how accurate the estimator will be when the dimension of the actions is high. [1] Policy Evaluation and Optimization with Continuous Treatments. [2] D4RL: Building Better Benchmarks for Offline Reinforcement Learning. ############## I have read the authors' response and other reviewers' comment, and I tend to keep my score.


Review 4

Summary and Contributions: This paper proposed several doubly robust off-policy value and policy gradient estimators for deterministic policy based on different kernelization approaches. Theoretical analysis show that the proposed estimator does not suffer from the curse of horizon even in the setting of H>=1. Finally, empirical results on a toy problem confirmed the theoretical results. =========After rebuttal…….. I have read the author feedback and would like to thank them for their responses to my questions. Although the authors respond that extensive experimentation is beyond the scope of this conference paper with so many new theoretical results already, I still feel that the paper should investigate more complex experimental tasks. Overall, this is a good paper and I will keep my score unchanged.

Strengths: The paper proposed several novel off-policy estimators for the policy evaluation and policy gradient with deterministic policy. The theoretical results are quite solid, and the topic is of great significance for the off-policy RL filed. The empirical results further demonstrate the effectiveness of the proposed estimator, and which are consist with the theoretical analysis.

Weaknesses: The expected return is a commonly measured performance in RL. Besides the MSE, is it possible to investigate the learning process and the final return performance of each estimator in the experimental part? The only concern I have is about the application. The chosen task is too simple, the implementation of the more complex real world problems are expected (as mentioned in section 6).

Correctness: To my knowledge, the method and empirical part are correct. I did not carefully check the proofs of the theoretical results.

Clarity: The paper is well-organized and well-written. Maybe it is better to review the most related work somehow, such as the kernelization approaches, since the technical details are difficult to follow. My comments are: 1. Add more intuition explanations for the Theorems; 2. It is better to explain more on the comparison results in Table 1. And, the introduction is not a suitable position for Table 1, since the proposed estimators are not introduced at that time.

Relation to Prior Work: The difference from related work is clearly discussed.

Reproducibility: Yes

Additional Feedback:

[Author Response · NeurIPS 2020]

We thank the reviewers for thoughtful reviews and encouraging comments. We respond only to questions and concerns.

(R1) "In the introduction ...": Good suggestion. Will refer specifically to CDRD, CDRK in Table 1a. The point is
extending the cited importance sampling methods to RL would lead to convergence rate that *deteriorates* in horizon.

(R1) "Something is said in Remark 2 ...": Our analysis handles known and unknown behavior policy and simply take
as a condition the nuisance estimation rate. Knowing the behavior policy can help estimate nuisances. In experiments
we consider known behavior policy as is common in offline RL. We still need to estimate $w_t^{\mathcal{K}}$ even if behavior is known.

(R1) "Out of the 9 combinations, ...": We focused on these as they represent the two extremes. We can easily provide in
the supplement a general analysis of all combinations under the intersection of the assumptions need for each extreme.

(R1) "In the beginning of section 4, ...": Unfortunately, no, as smoothness conditions are necessary in continuous
action space, else the finite data may be unrepresentative of the infinite possible unseen actions. This is the same as in
density estimation. We will comment on this.

(R1) "Although half of the paper ...": With limited space, we thought actual *learning* would be of greatest interest. We
will follow your suggestion and add to the supplement experiments for policy evaluation and varying $H$. Note this is
easy with the submitted code – we will just run it and report the results.

(R1) "The paper is well written ... In general, it would be interesting ...": The derivation is different and more
complicated than [3] as in [3] the density ratio exists and is used directly. On the other hand we need to analyze the
errors due to the kernelization, which complicates the analysis as it introduces slower leading terms with rate that
depends on horizon dimensionality.

(R1) "Typos: ... isn't the inverse?": Thanks for catching. Yes; the latter is a typo; it is the reverse.

(R2) "The experimental setting ...": The qualitative results are the same as we vary these. We will run the (submitted)
code with a range of parameters and include additional plots in supplement.

(R2) "The overall structure are well planned and the paper is well written ...": We will add reminders of notation when
used for first time much after problem setup and add short descriptions of steps in equations in appendix.

(R2) "Line 72–73 ...": Yes; it's a typo; "latter" and "former" should be exchanged.

(R2) "Line 81–82 ...": By Radon-Nikodym thm, exists $f$ such that $\mathbb{E}_{\pi_e}[g(a) \mid s] = \mathbb{E}_{\pi_b}[f(a)g(a) \mid s]$ for all $g$
measurable if and only if $\pi_e(\cdot \mid s)$ is absolutely continuous wrt $\pi_b(\cdot \mid s)$ (this is for each $s$). However, if $\pi_e(\cdot \mid s)$ is
discrete, it is ***not*** enough for behavior to have positive density at its atoms. E.g., Dirac at $0.5$ is ***not*** abs cts wrt the
uniform distribution on $[0,1]$ even though latter has density 1 at $0.5$. Recall $\mu \ll \nu$ means $(\mu(A) > 0 \Rightarrow \nu(A) >$
$0, \forall A$ measurable), so if $\mu \ll \nu$ and $\mu(\{0.5\}) = 1$ then $\nu(\{0.5\}) > 0$, i.e., has an atom at $0.5$. Will add this example.

(R3) "The novelty of ...": We respectfully disagree. Not only do we make it doubly robust *and* extend it to RL, we also
analyze it and give rates under lax conditions and show the naïve extension yields very bad rates as horizon grows.

(R3) "The evaluation is ...": Indeed while we avoid curse of dimension in state space and horizon, we may suffer
from it in action space, since we kernelize actions. In many practical offline RL settings, however, states are complex
(e.g., many+rich health indicators) and actions simple and often one-dimensional (e.g., insulin dosing/timing). We will
comment on this and run the (submitted) code on growing action dimension and add to supplement to visualize this.

(R4) "The expected return ...": In OPE, MSE is actually the metric of interest. In offline learning, policy value is
indeed of interest and our policy gradient experiments (Sec 5) showcase how low-error gradients lead to high-value
learning. Moreover, as Remark 8 mentions, error bound on gradients can be combined with standard gradient ascent
analysis to get value regret guarantees: we simply replace the stochastic-policy gradient error bounds of Kallus &
Uehara '20 with our new ones in Thms 11–13 therein; while this is straightforward use of existing work, we'll flesh this
out more explicitly in supplement for completeness.

(R4) "The only concern ...": The primary contribution is a theoretical study of rates and the simple experiments are
intended only to illustrate the new theory. Extensive experimentation is beyond the scope of such a short paper with so
many new results already. We can nonetheless easily run our (submitted) code on the Warfarin dosing experiment of
Kallus & Zhou '18 (the code is also public) and add to the supplement.

(R4) "The paper is well-organized and well-written ... more intuition explanations ... explain more on the comparison
results in Table 1": Thank you; we will use this feedback to further improve the clarity. We will add *more* in-words
explanations of each result and we will move Table 1.

[Meta-Review · NeurIPS 2020]

The paper is well written with solid theoretical contributions. Reviewers are also happy with the rebuttal. The common concern is that the paper lacks experimental evaluation.